# Repeated outbreaks drive the evolution of bacteriophage communication

**Hilje M Doekes[1,2]\*, Glenn A Mulder[1], Rutger Hermsen[1]**

[1]Theoretical Biology, Department of Biology, Utrecht University, Utrecht, Netherlands; [2]Laboratory of Genetics, Department of Plant Sciences, Wageningen University, Wageningen, Netherlands

**Abstract** Recently, a small-molecule communication mechanism was discovered in a range of *Bacillus*-infecting bacteriophages, which these temperate phages use to inform their lysis-lysogeny decision. We present a mathematical model of the ecological and evolutionary dynamics of such viral communication and show that a communication strategy in which phages use the lytic cycle early in an outbreak (when susceptible host cells are abundant) but switch to the lysogenic cycle later (when susceptible cells become scarce) is favoured over a bet-hedging strategy in which cells are lysogenised with constant probability. However, such phage communication can evolve only if phage-bacteria populations are regularly perturbed away from their equilibrium state, so that acute outbreaks of phage infections in pools of susceptible cells continue to occur. Our model then predicts the selection of phages that switch infection strategy when half of the available susceptible cells have been infected.

## Introduction

For several decades now, it has been recognised that communication between individuals is not limited to multicellular organisms, but is also common among microbes. The best-known example of microbial communication is bacterial *quorum sensing*, a process in which bacteria secrete signalling molecules to infer the local cell density and consequently coordinate the expression of certain genes (*Nealson et al., 1970*; *Miller and Bassler, 2001*). A wide variety of bacterial behaviours have been found to be under quorum-sensing control (*Miller and Bassler, 2001*; *Hense and Schuster, 2015*), including bioluminescence (*Nealson et al., 1970*), virulence (*Antunes et al., 2010*), cooperative public good production (*Diggle et al., 2007*; *Darch et al., 2012*), and antimicrobial toxin production (*Cornforth and Foster, 2013*; *Kleerebezem and Quadri, 2001*). Remarkably, it has recently been discovered that even some bacterial viruses (bacteriophages or phages for short) use signalling molecules to communicate (*Erez et al., 2017*). Here, we use a mathematical model to explore the dynamics of this viral small-molecule communication system. We study under what conditions communication between phages evolves and predict which communication strategies are then selected.

Bacteriophages of the SPbeta group, a genus in the order of *Caudovirales* of viruses that infect *Bacillus* bacteria, encode a small signalling peptide, named 'arbitrium', which is secreted when the phages infect bacteria (*Erez et al., 2017*). These phages are *temperate* viruses, meaning that each time a phage infects a bacterium, it makes a life-cycle decision: to enter either (i) the *lytic* cycle, inducing an active infection in which tens to thousands of new phage particles are produced and released through host-cell lysis, or (ii) the *lysogenic* cycle, inducing a latent infection in which the phage DNA is integrated in the host cell's genome (or episomally maintained) and the phage remains dormant until it is reactivated. This lysis-lysogeny decision is informed by the arbitrium produced in nearby previous infections: extracellular arbitrium is taken up by cells and inhibits the phage's lysogeny-inhibition factors, thus increasing the propenstiy towards lysogeny of subsequent infections (*Erez et al., 2017*). Hence, peptide communication is used to promote lysogeny when

\*For correspondence:
hiljedoekes@gmail.com

**Competing interests:** The authors declare that no competing interests exist.

**eLife digest** Bacteriophages, or phages for short, are viruses that need to infect bacteria to multiply. Once inside a cell, phages follow one of two strategies. They either start to replicate quickly, killing the host in the process; or they lay dormant, their genetic material slowly duplicating as the bacterium divides. These two strategies are respectively known as a 'lytic' or a 'lysogenic' infection.

In 2017, scientists discovered that, during infection, some phages produce a signalling molecule that influences the strategy other phages will use. Generally, a high concentration of the signal triggers lysogenic infection, while a low level prompts the lytic type. However, it is still unclear what advantages this communication system brings to the viruses, and how it has evolved.

Here, Doekes et al. used a mathematical model to explore how communication changes as phages infect a population of bacteria, rigorously testing earlier theories. The simulations showed that early in an outbreak, when only a few cells have yet been infected, the signalling molecule levels are low: lytic infections are therefore triggered and the phages quickly multiply, killing their hosts in the process. This is an advantageous strategy since many bacteria are available for the viruses to prey on. Later on, as more phages are being produced and available bacteria become few and far between, the levels of the signalling molecule increase. The viruses then switch to lysogenic infections, which allows them to survive dormant, inside their host.

Doekes et al. also discovered that this communication system only evolves if phages regularly cause large outbreaks in new, uninfected bacterial populations. From there, the model was able to predict that phages switch from lytic to lysogenic infections when about half the available bacteria have been infected.

As antibiotic resistance rises around the globe, phages are increasingly considered as a new way to fight off harmful bacteria. Deciphering the way these viruses communicate could help to understand how they could be harnessed to control the spread of bacteria.

many infections have occurred. Similar arbitrium-like systems have now been found in a range of different phages (*Stokar-Avihail et al., 2019*). Notably, these phages each use a slightly different signalling peptide and do not seem to respond to the signals of other phages (*Erez et al., 2017*; *Stokar-Avihail et al., 2019*).

The discovery of phage-encoded signalling peptides raises the question of how this viral communication system evolved. While the arbitrium system has not yet been studied theoretically, previous work has considered the evolution of lysogeny and of other phage-phage interactions. Early modelling work found that lysogeny can evolve as a survival mechanism for phages to overcome periods in which the density of susceptible cells is too low to sustain a lytic infection (*Stewart and Levin, 1984*; *Maslov and Sneppen, 2015*). In line with these model predictions, a combination of modelling and experimental work showed that selection pressures on phage virulence change over the course of an epidemic, favouring a virulent phage strain early on, when the density of susceptible cells is high, but a less virulent (i.e. lysogenic) phage strain later in the epidemic, when susceptible cells have become scarce (*Berngruber et al., 2013*; *Gandon, 2016*). Other modelling work has shown that if phages, lysogenised cells, and susceptible cells coexist for long periods of time, less and less virulent phages are selected (*Mittler, 1996*; *Wahl et al., 2018*). This happens because phage exploitation leads to a low susceptible cell density, and hence a virulent strategy in which phages rapidly lyse their host cell to release new phage particles that can then infect other cells no longer pays off (because few cells are available to infect).

*Erez et al., 2017* propose that the arbitrium system may have evolved to allow phages to cope with the changing environment during an epidemic, allowing the phages to exploit available susceptible bacteria through the lytic cycle when few infections have so far taken place and hence the concentration of arbitrium is low, while entering the lysogenic cycle when many infections have taken place and the arbitrium concentration has hence increased. This explanation resembles results for other forms of phage-phage interaction previously found in *Escherichia coli*-infecting phages (*Abedon, 2017*; *Abedon, 2019*). In the obligately lytic T-even phages, both the length of the latent period of an infection and the subsequent burst size increase if additional phages adsorb to the cell

while it is infected – a process called *lysis inhibition* (*Hershey, 1946*; *Doermann, 1948*; *Abedon, 2019*). In the temperate phage λ, the propensity towards lysogeny increases with the number of co-infecting virions, called the multiplicity of infection (MOI) (*Kourilsky, 1973*). In both cases, modelling work has shown that the effect of the number of phage adsorptions on an infection can be selected as a phage adaptation to host-cell density, as it allows phages to switch from a virulent infection strategy (i.e. a short latent period or a low lysogeny propensity) when the phage:host-cell ratio is low to a less virulent strategy (i.e. a longer latent period or higher lysogeny propensity) when the phage:host-cell ratio is high (*Abedon, 1989*; *Abedon, 1990*; *Sinha et al., 2017*).

Here, we present a mathematical model to test if similar arguments can explain the evolution of small-molecule communication between viruses, and to explore the ecological and evolutionary dynamics of temperate phage populations that use such communication systems. We show that arbitrium communication can indeed evolve and that communicating phages consistently outcompete phages with non-communicating bet-hedging strategies. We however find that communication evolves under certain conditions only, namely if the phages regularly cause new outbreaks in substantial pools of susceptible host cells. Moreover, when communication evolves under such conditions, we predict that a communication strategy is selected in which phages use arbitrium to switch from a fully lytic to a fully lysogenic strategy when approximately half of all susceptible cells have been infected. Finally, we investigate how reliable the arbitrium signal needs to be for such communication to evolve, and show that the results are remarkably robust against variation in the density of bacteria.

## Materials and methods

### Model

Following earlier models (e.g. *Stewart and Levin, 1984*; *Berngruber et al., 2013*; *Sinha et al., 2017*; *Wahl et al., 2018*), we use ordinary differential equations to describe a well-mixed system consisting of susceptible bacteria, lysogens (i.e. lysogenically infected bacteria), and free phages, but extend this system to include an arbitrium-like signalling peptide (*Figure 1A*). For simplicity, we consider phages that do not affect the growth of lysogenised host cells; susceptible bacteria and lysogens hence both grow logistically with the same growth rate $r$ and carrying capacity $K$. Lysogens are spontaneously induced at a low rate $\alpha$, after which they lyse and release a burst of $B$ free phages per lysing cell. Free phage particles decay at a rate $\delta$ and adsorb to bacteria at a rate $a$. Adsorptions to lysogens result in the decay of the infecting phage, thus describing the well-known effect of superinfection immunity (*Hutchison and Sinsheimer, 1971*; *Susskind et al., 1974*; *McAllister and Barrett, 1977*; *Kliem and Dreiseikelmann, 1989*; *Bondy-Denomy et al., 2016*), whereas adsorptions to susceptible bacteria result in infections with success probability $b$. We consider the lytic cycle to be fast compared to both bacterial growth and the lysogenic cycle (*Stewart and Levin, 1984*; *Berngruber et al., 2013*; *Sinha et al., 2017*; *Wahl et al., 2018*), so that a lytic infection can be modelled as immediate lysis releasing a burst of $B$ free phages. Since the genes encoding arbitrium production are among the first genes to be expressed when a phage infects a host cell (*Erez et al., 2017*; *Stokar-Avihail et al., 2019*), each infection leads to an immediate increase of the arbitrium concentration $A$ by an increment $c$. The lysis-lysogeny decision is effected by the current arbitrium concentration: a fraction $\varphi(A)$ of the infections results in the production of a lysogen, while the remaining fraction $(1 - \varphi(A))$ results in a lytic infection. Arbitrium does not decay spontaneously in the model (since it is a small peptide, spontaneous extracellular degradation is considered to be negligible), but it is taken up by bacteria at a rate $u$ (e.g. through general bacterial peptide importers such as OPP [*Erez et al., 2017*]), and then degraded intracellularly, thus reducing the arbitrium concentration $A$.

Consider competing phage variants $i$ that differ in their (arbitrium-dependent) lysogeny propensity $\varphi_i(A)$. The population densities (cells or phages per volume unit) of susceptible bacteria $S$, phage particles $P_i$ and corresponding lysogens $L_i$, and the concentration of arbitrium $A$ can then be described by:

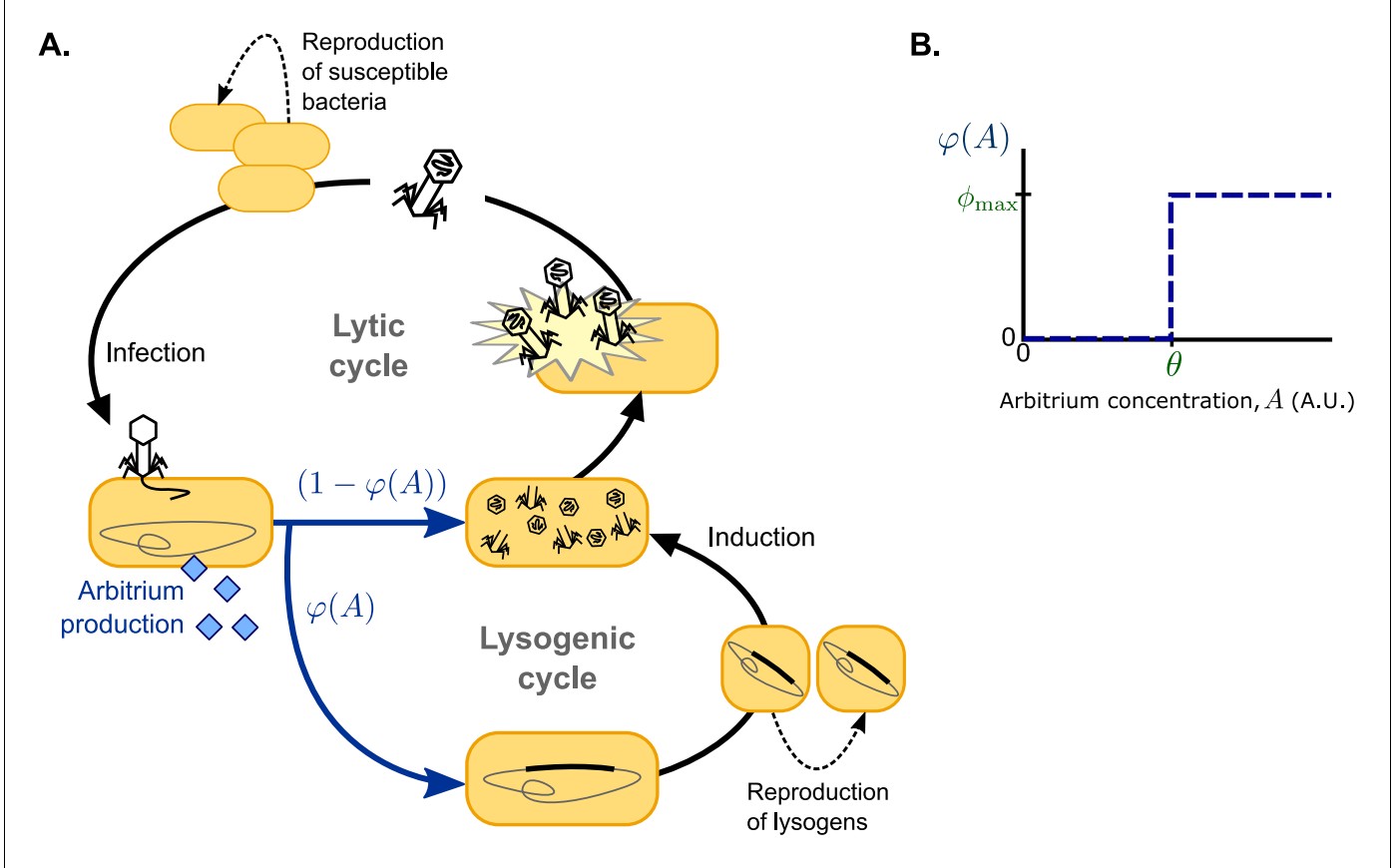

**Figure 1.** Model overview. (**A**) Free phages infect susceptible bacteria, at which point a fixed amount of arbitrium is produced. This arbitrium is taken up and degraded by susceptible cells and lysogens. Upon infection, a cell enters the lysogenic cycle with propensity $\varphi(A)$, or the lytic cycle with propensity $(1 - \varphi(A))$; the lysogeny propensity $\varphi(A)$ depends on the current arbitrium concentration. The lytic cycle leads to immediate lysis of the host cell and release of a burst of new virions. In the lysogenic cycle, the phage remains dormant in the lysogen population, which grows logistically with the same rate as the susceptible cell population. Lysogens are spontaneously induced at a low rate, at which point they re-enter the lytic cycle. (**B**) In communicating phages, the lysogeny propensity $\varphi(A)$ is modelled by a step-function characterised by two phage characteristics: $\theta$, the arbitrium concentration above which the phage increases its lysogeny propensity, and $\phi_{\max}$, the lysogeny propensity of the phage at high arbitrium concentration.

$$\frac{\mathrm{d}S}{\mathrm{d}t} = \underbrace{rS(1 - N/K)}_{\text{logistic growth}} - \underbrace{baS\sum_i P_i}_{\text{infection}}, \tag{1}$$

$$\frac{\mathrm{d}L_i}{\mathrm{d}t} = \underbrace{rL_i(1 - N/K)}_{\text{logistic growth}} + \underbrace{\varphi_i(A)baSP_i}_{\text{lysogenic infection}} - \underbrace{\alpha L_i}_{\text{induction}}, \tag{2}$$

$$\frac{\mathrm{d}P_i}{\mathrm{d}t} = \underbrace{B\alpha L_i}_{\text{burst from induction}} + \underbrace{B(1 - \varphi_i(A))baSP_i}_{\text{burst from lytic infection}} - \underbrace{\delta P_i}_{\text{phage decay}} - \underbrace{aNP_i}_{\text{adsorption}}, \tag{3}$$

$$\frac{\mathrm{d}A}{\mathrm{d}t} = \underbrace{cbaS\sum_i P_i}_{\text{production upon infection}} - \underbrace{uNA}_{\text{adsorption and degradation}}, \tag{4}$$

where $N = S + \sum_i L_i$ is the total density of bacteria.

We study two scenarios for the lysis-lysogeny decision: (i) a baseline scenario in which the arbitrium concentration does not affect the lysis-lysogeny decision; each phage variant has a constant lysogeny propensity $\phi_i$ and (ii) a full scenario in which the arbitrium concentration does affect the lysis-lysogeny decision; each phage variant causes lytic infection when the arbitrium concentration is

low, but switches to some lysogeny propensity $\phi_{\mathrm{max}_i}$ when the arbitrium concentration exceeds the phage's response threshold $\theta_i$ (*Figure 1B*; note that we use $\phi_{\mathrm{max}_i}$ to denote a constant characteristic of the phage and $\varphi_i$ to denote the function describing how phage variant $i$'s lysogeny propensity depends on the arbitrium concentration). In this second scenario, phage variants with constant lysogeny propensity are still included: variants with a response threshold $\theta_i = 0$ cause lysogenic infections with lysogeny propensity $\phi_{\mathrm{max}_i}$ independent of the arbitrium concentration. Scenario (ii) is hence an extension of scenario (i).

On top of the ecological processes described in *Equation 1–4*, the model also allows for evolution of the phages due to mutations that change the characteristics $\phi_{(\mathrm{max})}$ and $\theta$ of phage variants. In *Equation 1–4* terms describing these mutations were omitted for readability; they are described in detail in Appendix A1.1. In short, replication of any phage variant was assumed to produce mutants with slightly different characteristics (e.g. a slightly higher or lower lysogeny propensity) with a small probability μ. Under scenario (ii), mutations changing $\phi_{\mathrm{max}}$ and $\theta$ are implemented as independent processes.

## Serial passaging

In natural settings as well as in some laboratory experiments, phages regularly cause large outbreaks in pools of susceptible cells that were previously unavailable to the phages (e.g. when phages are spread to a new area, or when phages are serially passaged in a lab setting). Such outbreaks perturb the phage and cell populations away from their equilibrium. To mimic such repeated perturbations, we expose the system of *Equation 1–4* to a phage serial-passaging regime (mimicking the experimental set-up of, for example, *Bull et al., 1993*; *Bull et al., 2004*; *Bollback and Huelsenbeck, 2007*; *Betts et al., 2013*; *Broniewski et al., 2020*). We initialise the model with a susceptible bacterial population at carrying capacity ($S = K$ cells per mL) and a small phage population ($\sum_i P_i = 10^6$ phages per mL) and numerically integrate *Equation 1–4* for a time of $T$ hours. Then a fraction of the phage population is taken and transferred to a new population of susceptible bacteria at carrying capacity and *Equation 1–4* are again integrated for $T$ hours. This cycle is repeated to bring about a long series of epidemics. Throughout the manuscript, a dilution factor of $D = 0.01$ is used (i.e. the passaged sample is 1% of the phage population). Passaging does not alter the relative frequency of the different phage variants, thus ensuring that the phage variants that were highly prevalent in the phage population at the end of an episode remain at a high relative frequency at the start of the new episode.

In this set-up, only phages are passaged from one epidemic episode to the next. To assess the robustness of simulation results to changes in this protocol, a second set-up was considered in which a fraction of the *full* sample (susceptible cells, lysogens, phages, and arbitrium) was passaged. We furthermore tested how the results are affected by variation in the bacterial carrying capacity. For this, at the start of each episode a carrying capacity value was sampled from a gamma distribution with mean $K$. We control the level of noise through the variance of this gamma distribution.

## Parameters

In total, the model (*Equation 1–4*) has nine parameters (excluding the phage characteristics $\phi_i$, $\phi_{\mathrm{max}_i}$, and $\theta_i$, which vary between phage variants present in any given simulation). As far as we are aware, none of these have been estimated for phages of the SPBeta group, but many have been measured for other phages, most of which infect *E. coli* (*Table 1*, estimates taken from *Little et al., 1999*; *De Paepe and Taddei, 2006*; *Wang, 2006*; *Shao and Wang, 2008*; *Zong et al., 2010*; *Berngruber et al., 2013*). To reduce the number of parameters in our analysis, we nondimensionalised the equations to obtain five scaled parameter values (Appendix A1.3) and used the literature estimates to derive default values for these scaled parameters (*Table 1*). To account for the uncertainty in these estimates, we performed parameter sweeps consisting of 500 simulations with parameter values randomly sampled from broad parameter ranges (*Table 1*). To ensure that low values of the parameters were well-represented, parameter values were sampled log-uniformly.

## Model analysis

Numerical integration was performed in Matlab R2017b, using the default built-in ODE-solver `ode45`. Scripts are available from https://github.com/hiljedoekes/PhageCom.

**Table 1.** Model parameters.

| | | | | |
|---|---|---|---|---|
| | | **Original parameters** | | |
| Parameter | Description (dimension) | Literature estimates | References | |
| $r$ | Net replication rate of bacteria (hour$^{-1}$) | 1.0 | *Berngruber et al., 2013* | |
| $K$ | Carrying capacity of bacteria (cells mL$^{-1}$) | $10^9$ | *Berngruber et al., 2013* | |
| $a$ | Adsorption rate of phages to bacteria (hour$^{-1}$ (cells per mL)$^{-1}$) | $10^{-9}$— $10^{-7}$ | *De Paepe and Taddei, 2006*; *Shao and Wang, 2008* | |
| $b$ | Proportion of adsorptions of a phage to a susceptible cell that leads to infection (cells phage$^{-1}$) | set at $10^{-2}$, not measured | *Berngruber et al., 2013* | |
| $B$ | Burst size (phages) | 10— $3.5 \cdot 10^3$ | *De Paepe and Taddei, 2006*; *Wang, 2006* | |
| $\alpha$ | Rate of spontaneous lysogen induction (hour$^{-1}$) | $10^{-4}$—$10^{-3}$ | *Little et al., 1999*; *Zong et al., 2010*; *Berngruber et al., 2013* | |
| $\delta$ | Spontaneous decay rate of free phages (hour$^{-1}$) | $10^{-3}$—$2 \cdot 10^{-2}$ | *De Paepe and Taddei, 2006* | |
| $u$ | Uptake rate of arbitrium by cells (arbitrium mL$^{-1}$ (cells per mL)$^{-1}$) | no estimates known | - | |
| | | **Scaled dimensionless parameters used in parameter sweeps** | | |
| Parameter | Description | Default value | Parameter sweep range | |
| $\hat{B} = bB$ | Effective burst size | 2 | 1—$10^3$ | |
| $\hat{a} = \frac{aK}{r}$ | Scaled adsorption rate of phages to cells | 10 | 1—100 | |
| $\hat{\delta} = \frac{\delta}{r}$ | Scaled decay rate of phage particles | 0.01 | $10^{-3}$—0.1 | |
| $\hat{\alpha} = \frac{\alpha}{r}$ | Scaled spontaneous phage induction rate | $10^{-3}$ | $10^{-4}$—$10^{-2}$ | |
| $\hat{u} = \frac{uK}{r}$ | Scaled rate of arbitrium uptake and degradation by cells | 0.1 | $10^{-3}$—1 | |
| $D$ | Dilution factor of phages at serial passages | 0.01 | $10^{-3}$—0.1 | |

Next to numerical integration results, we analytically found expressions for the model equilibria and derived expressions for the evolutionarily stable strategy (ESS) under serial passaging in both scenarios (excluding and including arbitrium communication). Detailed derivations are provided in Appendix A3.

## Results

### Evolution of the lysis-lysogeny decision and arbitrium communication requires perturbations away from equilibrium

A common approach to analysing ODE-models such as *Equation 1–4* is to characterise the model's equilibrium states (*Stewart and Levin, 1984*; *Wahl et al., 2018*; *Cortes et al., 2019*). Such an analysis is provided in Appendix A2. However, we will here argue that to understand the evolution of arbitrium communication, and the lysis-lysogeny decision in general, considering the equilibrium states is insufficient.

Firstly, the function of the arbitrium system is to allow phages to respond to changes in the density of susceptible cells and phages as reflected in the arbitrium concentration. But when the system approaches an equilibrium state, the densities of susceptible cells and phages become constant, and so does the arbitrium concentration. Equilibrium conditions hence defeat the purpose of small-molecule communication such as the arbitrium system. Evolution of small-molecule communication must be driven by dynamical ecological processes, and hence can only be studied in populations that are regularly perturbed away from their ecological steady state.

Secondly, under equilibrium conditions natural selection can act on the lysis-lysogeny decision only if infections still take place, and hence lysis-lysogeny decisions are still taken. We argue that this is unlikely. If the phage population is viable (i.e. if the parameter values are such that the phages proliferate when introduced into a fully susceptible host population), the model converges to one of two qualitatively different equilibria, depending on parameter conditions (Appendix A2): either (i)

susceptible host cells, lysogens and free phages all coexist, or (ii) all susceptible host cells have been infected so that only lysogens and free phages remain. The evolution of a constant lysogeny propensity in a host-phage population with a stable equilibrium of type (i) was recently addressed by Wahl et al., who show that under these conditions selection always favours phage variants with high lysogeny propensity (i.e. $\phi = 1$) (*Wahl et al., 2018*). However, only a narrow sliver of parameter conditions permits a stable equilibrium of type (i) (*Sinha et al., 2017*; *Cortes et al., 2019*), and when we estimated reasonable parameter conditions based on a variety of well-studied phages, we found that they typically lead to a stable equilibrium of type (ii) (Appendix A2, parameter estimates based on *Little et al., 1999*; *De Paepe and Taddei, 2006*; *Wang, 2006*; *Shao and Wang, 2008*; *Zong et al., 2010*; *Berngruber et al., 2013*). This is because phage infections tend to be highly effective: their large burst size and consequent high infectivity cause temperate phages to completely deplete susceptible host cell populations, replacing them with lysogens that are immune to superinfection and hence have a strong competitive advantage over the susceptible cells (*Bossi et al., 2003*; *Gama et al., 2013*). Under common parameter conditions, after a short epidemic the susceptible cells population is hence depleted and no more infections take place, causing the competition between different phage variants to cease (see *Figure 2A* for example dynamics). Then there is no long-term selection on the lysis-lysogeny decision, and studying its evolution in this state is pointless.

We therefore consider a scenario in which the phage and cell populations are regularly perturbed away from equilibrium. To do so, we simulate serial-passaging experiments by periodically transferring a small fraction of the phages to a new population of susceptible host cells at carrying capacity, thus simulating cycles of repeated outbreaks (see Materials and methods).

## In the absence of arbitrium communication, bet-hedging phages are selected with low constant lysogeny propensity

To form a baseline expectation of the evolution of the lysis-lysogeny decision under the serial-passaging regime, we first considered a population of phage variants that do not engage in arbitrium communication, but do differ in their constant lysogeny propensity $\phi_i$. Under typical parameter conditions (default values in *Table 1*), each passaging episode starts with an epidemic in which the susceptible cell population is depleted, followed by a period in which the bacterial population is made up of lysogens only (*Figure 2A*, dynamics shown for a passaging episode length $T = 12$ h). The composition of the phage and lysogen populations initially changes over subsequent passaging episodes (*Figure 2A*), but eventually an evolutionarily steady state is reached in which one phage variant dominates the phage population ($\phi = 0.04$; *Figure 2B*), confirming that the lysis-lysogeny decision is indeed under selection.

The distribution of phage variants at evolutionarily steady state depends on the time between passages, $T$ (*Figure 2C*). If this time is short ($T \leq 5$ h), the phage variant with $\phi = 0$ dominates at evolutionarily steady state. This is an intuitive result: under these conditions phages are mostly exposed to environments with a high density of susceptible cells, in which a lytic strategy is favourable. Surprisingly, however, if the time between passages is sufficiently long ($T > 5$ h), the viral population at evolutionarily steady state always centres around the same phage variant, independent of $T$ ($\phi = 0.04$; *Figure 2C*). This result can be explained by considering the dynamics within a passaging episode (see *Figure 2A*): Once the susceptible cell population has collapsed, free phages no longer cause new infections and are hence 'dead ends'. New phage particles are then formed by reactivation of lysogens only, so that the distribution of variants among the free phages comes to reflect the relative variant frequencies in the lysogen population. Hence, when the time between passages is sufficiently long, the phage type that is most frequent in the sample that is eventually passaged is the one that is most frequent in the population of lysogens (*Figure 2—figure supplement 1*). Under default parameter conditions, these are the phages with a low lysogeny propensity of $\phi = 0.04$.

Note that although a single phage variant clearly dominates the population, some diversity is maintained (*Figure 2B–C*). This is due to a mutation-selection balance: because mutants with slightly different $\phi$-values continuously arise from the dominant phage variant and selection against these mutants is weak (due to their similarity to the dominant phage variant), the balance between influx of mutant variants by mutation and their efflux by selection results in the long-term presence of these mutants in the population. Such a population consisting of a dominant variant and its close mutants is called a quasi-species (*Eigen, 1971*).

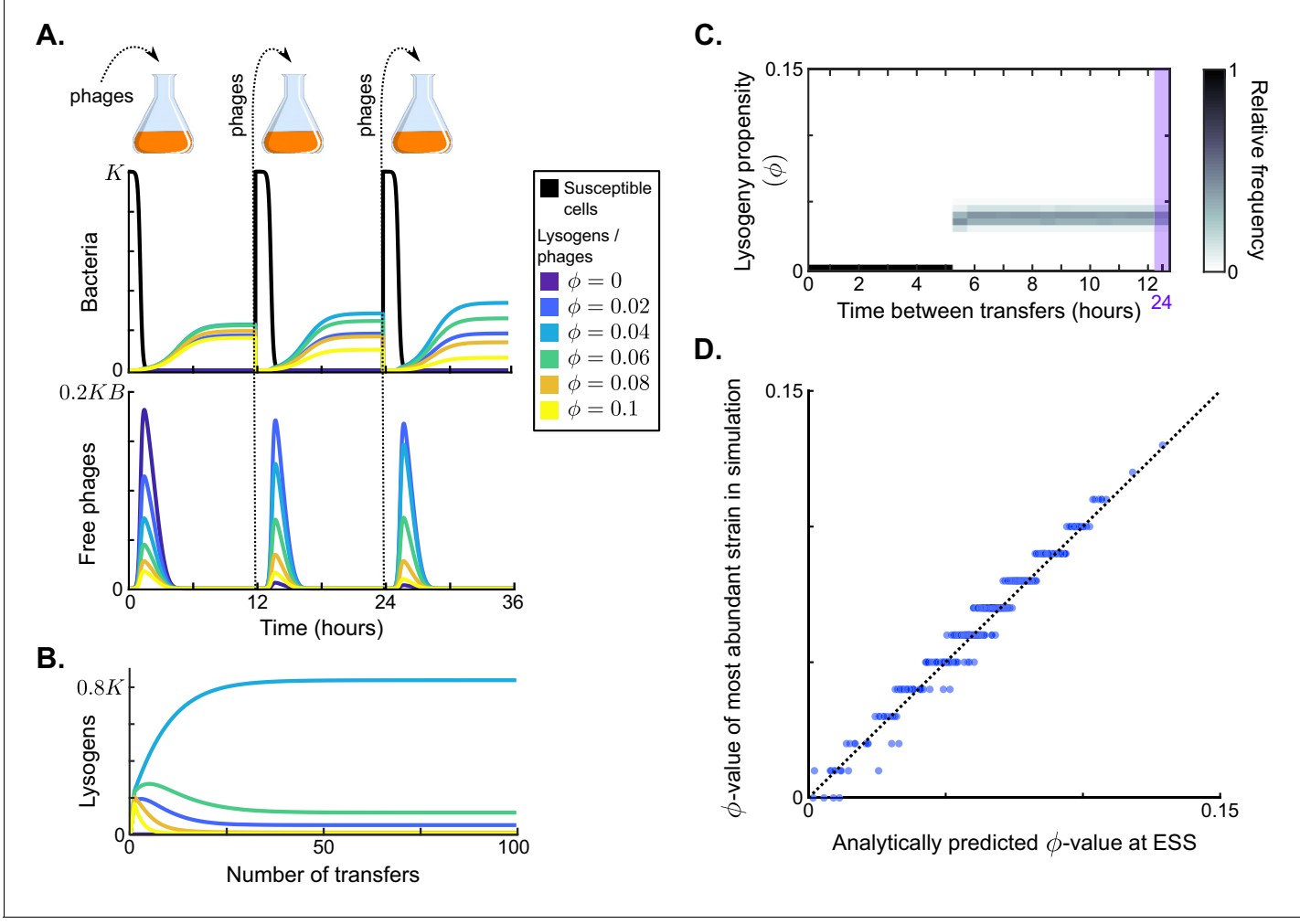

**Figure 2.** Results in the absence of phage communication. (A) Short-term model dynamics under default parameter conditions (*Table 1*) and a passaging episode duration of $T = 12$ h. The model was initialised with a susceptible bacterial population at carrying capacity ($S = K$) and a low frequency of phages ($\sum_i P_i = 10^{-5}KB$), and upon passaging the phages were diluted 100-fold. Phage variants differ in their lysogeny propensity $\phi_i$. Dynamics within a single passaging episode are further illustrated in *Figure 2—figure supplement 1*. (B) Long-term model dynamics for default parameter settings and $T = 12$ h. Over many passages, a single phage variant ($\phi = 0.04$) is selected. (C) Distribution of phage variants at evolutionarily steady state as a function of the time between passages, $T$. A total number of 101 phage variants was included, with lysogeny propensities varying between $\phi_1 = 0$ and $\phi_{101} = 0.5$. When the interval between passages is short, the susceptible cells are not depleted during the rounds of infection and a fully lytic strategy ($\phi = 0$) is selected. For larger values of $T$, however, a bet-hedging strategy with small but non-zero $\phi$-value is selected ($\phi \approx 0.04$). Almost identical results are obtained if the serial passaging set-up is altered to simulate serial passaging of a full sample (phages, susceptible bacteria, and lysogens) instead of phages only (*Figure 2—figure supplement 2*). (D) Parameter sweep results. The model was run with 500 sets of randomly sampled parameters, and for each run the most abundant $\phi$-value in the population at evolutionarily steady state was plotted against the analytically predicted evolutionarily stable strategy (ESS; see Appendix A3 and *Box 1*). The dotted line is the identity line. The analytically derived ESS is a good predictor of the simulation outcome.

The online version of this article includes the following figure supplement(s) for figure 2:

**Figure supplement 1.** Model dynamics during a single passaging episode for default parameter values and constant lysogeny propensities (no communication).

**Figure supplement 2.** Distribution of phage variants at evolutionarily steady state as a function of the time between passages, $T$, for constant lysogeny propensities (no communication) under a serial-passaging regime in which full samples (susceptible bacteria, lysogens, and phages) are passaged.

Next, we assessed the robustness of the results to changes in the serial passaging protocol. In the standard protocol, only phages are passaged between episodes. If instead the passaged sample consists of the full system (susceptible cells, lysogens, and phages), almost identical results are obtained (*Figure 2—figure supplement 2*). This is again explained by realising that, as long as the

time between passages is sufficiently long, the distribution of variants in the free phages is equal to the distribution in the lysogens. Since lysogens need to be induced to contribute to a new outbreak and the induction rate $\alpha$ is the same for all phage variants, the contribution of passaged lysogens to the new outbreak does not alter the relative frequency of phage variants.

To examine how these results depend on the model parameters, we determined which phage variant was most abundant at evolutionarily steady state for 500 randomly chosen parameter sets (see *Table 1* for parameter ranges), always using a long time between passages ($T = 24$ h). The selected $\phi$-values for all parameter settings lie between $\phi = 0$ and $\phi = 0.12$ (*y*-axis of *Figure 2D*). We can hence conclude that selection favours phages with low but usually non-zero lysogeny propensities. These phages employ a bet-hedging strategy: throughout the epidemic they 'invest' a small part of their infection events in the production of lysogens, such that they are maximally represented in the eventual lysogen population.

To better understand how the lysogeny propensity $\phi$ that is selected depends on parameter values, we derived an analytical approximation for the evolutionarily stable strategy (ESS) under the serial-passaging regime if the time between passages is sufficiently long (Appendix A3.1–2). Because the phage dynamics during an epidemic affect the dynamics of the susceptible cells and vice versa, phage fitness is frequency dependent and the ESS is not found by a simple optimisation procedure, but by identifying the particular $\phi$-value, denoted $\phi^*$, that maximises phage fitness given that this strategy $\phi^*$ itself shapes the dynamics of the epidemic (*Box 1*). We find that the ESS can be approximated by the surprisingly simple expression

$$\phi^* = \frac{1 - (bB)^{-1}}{\log\left(\frac{BK}{P_0}\right)}, \tag{5}$$

where $P_0$ is the density of phages at the start of a passaging episode. This approximation corresponds well with the results of the parameter sweep (*Figure 2D*), indicating that it indeed captures the most important factors shaping the evolution of the lysogeny propensity $\phi$.

*Equation 5* shows that the ESS depends on the initial phage density in a passaging episode, $P_0$, relative to the burst size $B$ and maximal host-cell density $K$, and the effective burst size $bB$, which represents the expected number of progeny phages per phage that adsorbs to a susceptible bacterium. The ESS $\phi^*$ decreases with the dilution factor of the phages upon passage (i.e. with lower $P_0$). On the other hand, $\phi^*$ increases with the effective burst size $bB$ (note that $(bB)^{-1}$ decreases when $(bB)$ increases). Both effects can be intuitively understood by considering how these factors affect the duration of the epidemic, $T_E$. If the phage density is low at the start of a passaging episode or if the phages have a small effective burst size, it takes a while before the phage population has grown sufficiently to cause the susceptible population to collapse. Since a lytic strategy is favoured early in the epidemic, when the susceptible cell density is still high, a longer epidemic favours phages with lower values of $\phi$ (see the red line in the figure in *Box 1*). On the other hand, if the initial phage density is high or if the phages have a high effective burst size, the susceptible cell population collapses quickly, phages have a much shorter window of opportunity for lysogen production and hence phages with higher $\phi$-values are favoured.

## If arbitrium communication is included, communicating phages are selected that switch from a fully lytic to a fully lysogenic strategy

Next, we included the possibility of arbitrium communication and let phage variants be characterised by two properties: their arbitrium response threshold, $\theta_i$, and their lysogeny propensity when the arbitrium concentration exceeds their response threshold, $\phi_{\max_i}$ (see *Figure 1B*). We then again considered the dynamics of our model under a serial-passaging regime.

In *Figure 3A*, example dynamics are shown for three competing phage variants, all with $\phi_{\max} = 1$ but with different response thresholds $\theta_i$. The arbitrium concentration increases over the course of the epidemic, and the phage variants switch from lytic infection to lysogen production at different times because of their different response thresholds.

Note that the maximum arbitrium concentration obtained during a passaging episode is approximately $A = cK$ (*Figure 3A*). This is because during the epidemic the dynamics of the susceptible cell density are mostly determined by infection events and not so much by the (slower) bacterial growth.

## Box 1. Lysogeny propensity of the evolutionarily stable strategy (ESS).

An evolutionarily stable strategy (ESS) is a strategy that cannot be invaded by any other strategy. In the context of the lysogeny propensity $\phi$, it is the value $\phi^*$ such that a population currently dominated by a phage with $\phi = \phi^*$ cannot be invaded by any phage variant with a different $\phi$-value. A phage variant with $\phi = \phi_i$ invading in a resident population with the same $\phi = \phi_i$ always grows exactly like the resident. If this is the best possible invader, any other phage variant must perform worse than the resident and cannot invade. Hence, the ESS is the *optimal response to itself*. However, we still have to define what it means to be the ''best possible invader'' under the serial-passaging regime. Note that if the time between passages is sufficiently long, phages are selected on their ability to produce lysogens during the active epidemic (see Main Text). The optimal invader is hence the phage variant that, when introduced at a very low frequency, produces the most lysogens *per capita* between time $t = 0$ and the time that the susceptible cell population collapses, $T_E$. The $\phi$-value of the optimal invader depends on $T_E$ (red line in plot): if the epidemic phase is short, lysogens have to be produced quickly and a high $\phi$-value is optimal, while if the epidemic lasts longer, phages can profit more from lytic replication and a lower $\phi$-value is optimal. In turn, however, the duration of the epidemic $T_E$ depends on the lysogeny propensity $\phi$ of the resident phage population (blue line in plot): phages with a lower value of $\phi$ replicate more rapidly and hence cause an earlier collapse of the susceptible population. The ESS is the value $\phi^*$ that is optimal given the collapse time $T_E(\phi^*)$ that results when $\phi^*$ itself is the resident strategy. Graphically, this value can be identified as the intersection of $T_E(\phi)$ and $\phi_{\mathrm{opt}}(T_E)$ (the red and blue lines).

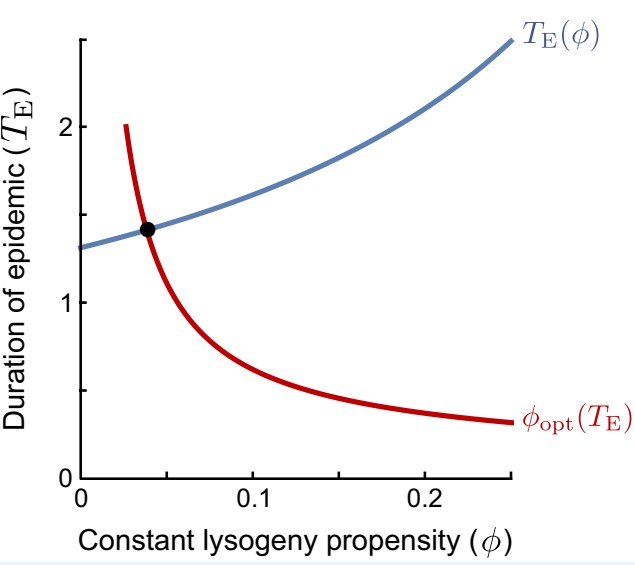

**Box 1—figure 1.** The ESS is found as the intersection of the curves of (i) the duration of the epidemic as a function of the lysogeny propensity of the resident (blue line), and (ii) the optimal lysogeny propensity of the invader given a fixed duration of the epidemic (red line).

Since the arbitrium concentration increases by an increment $c$ every time a susceptible cell is infected, the infection of all initial susceptible cells will result in an arbitrium concentration of $A = cK$ (assuming that the degradation of arbitrium is also slow and can be ignored during the growth phase of the epidemic). The arbitrium concentration during the early epidemic then is a direct reflection of the fraction of susceptible cells that have so far been infected.

To study the evolution of arbitrium communication, we again considered the distribution of phage variants at evolutionary steady state for varying values of the time between passages, $T$. Similar to the results shown in *Figure 2C*, we find two main regimes (*Figure 3B*): if the time between passages is short ($T<4$ h, illustrated by $T = 2$ h in *Figure 3B*), selection favours phage variants that only cause lytic infections ($\phi_{\max} = 0$); if the time between passages is sufficiently long ($T \geq 5$ h, illustrated by $T = 12$ h in *Figure 3B*), the phage population is dominated by variants with $\phi_{\max} = 1$ and $\theta \approx 0.65cK$. For $4 \leq T<5$ h, we see a transition between these two regimes (*Figure 3—figure supplement 1*). If the time between passages is sufficiently long ($T>5$ h), phage variants are hence selected that switch from a completely lytic to a completely lysogenic strategy when the arbitrium concentration exceeds a certain threshold.

In the simulations of *Figure 3B*, phage variants could have emerged that use the bet-hedging strategy found in the absence of communication (in phage variants with $\theta = 0$, the lysogeny propensity is always $\phi_{\max}$, independently of the arbitrium concentration), but this did not happen. We can hence conclude that any bet-hedging phage variants were outcompeted by variants that do use arbitrium communication. To underscore this conclusion, we simulated a competition experiment between the bet-hedging phage variant that was selected in the absence of communication and the communicating variant selected when arbitrium dynamics were included (*Figure 3C*). The communicating phage quickly invades on a population of bet-hedging phages and takes over, confirming that communication is indeed favoured over bet-hedging.

If a full sample (susceptible cells, lysogens, phages, and arbitrium) is passaged instead of phages only, again almost identical results are found (*Figure 3—figure supplement 2*). As was the case for the simulations in which arbitrium was absent, passaged lysogens do not alter the distribution of phage variants in the new outbreak. The passaged arbitrium does not significantly affect the outbreak dynamics either, because its concentration after dilution is much lower than the response threshold θ of the phage variants that are selected.

## Evolved phages switch from the lytic to the lysogenic life-cycle when approximately half of the susceptible cells have been infected

To study how the evolution of phage communication depends on phage and bacterial characteristics, 500 simulations were performed with randomly sampled sets of parameter values (*Table 1*), using a long time between serial passages ($T = 24$ h). For each simulation, we determined which phage variant was most prevalent at evolutionary steady state. Although we varied the parameter values over several orders of magnitude, the most prevalent phage variant had a lysogeny propensity of $\phi_{\max} = 1$ and a response threshold of $\theta = 0.5cK$ or $\theta = 0.6cK$ in almost all simulations (*Figure 4A*). Hence, over a broad range of parameter values, phages are selected that use the arbitrium system to switch from a fully lytic to a fully lysogenic strategy (i.e. $\phi_{\max} = 1$). This suggests that over the course of an epidemic, there is an initial phase during which the lytic strategy is a 'better' choice (i.e. produces the most progeny on the long run), while later in the epidemic the production of lysogens is favoured and residual lytic infections that would results from a lysogeny propensity $\varphi<1$ are selected against.

To better understand the intriguing consistency in θ-values found in the parameter sweep, we used a similar approach as before to analytically derive an approximation for the response threshold $\theta^*$ of the evolutionarily stable strategy under the condition that the time between passages is long (Appendix A3.3). Again, we find a surprisingly simple expression for the ESS:

$$\theta^* = \frac{cK}{2 - (bB)^{-1}}. \tag{6}$$

Note that the expression in *Equation 6* again depends on the effective burst size $bB$, which is an indicator of the phage's infectivity. The evolutionarily stable response threshold $\theta^*$ declines as the effective burst size increases, converging to a value of $\theta^* = \frac{1}{2}cK$ for highly infective phages

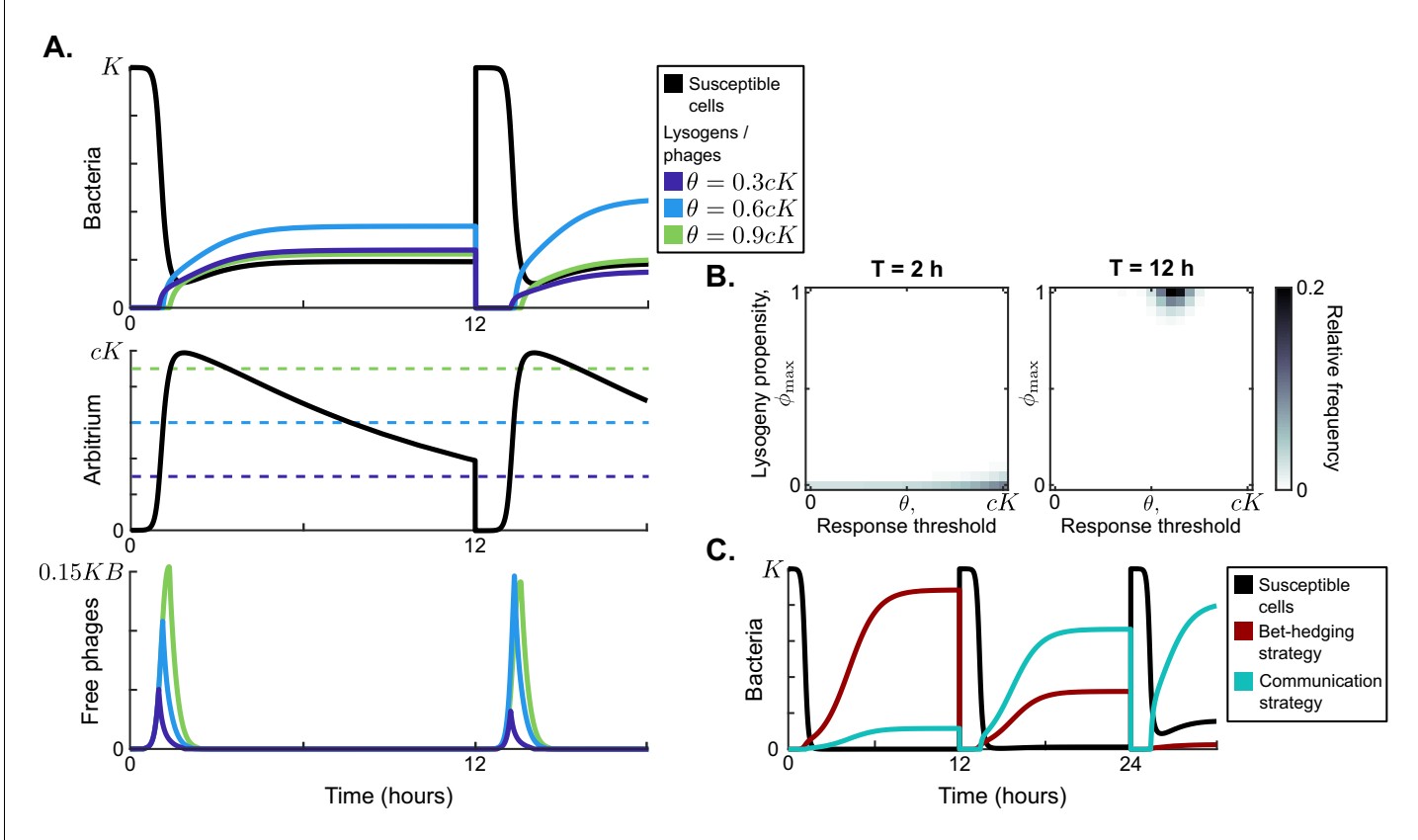

**Figure 3.** Model dynamics if phage communication is included in the model. (A) Short-term dynamics for default parameter conditions (*Table 1*) and the same serial-passaging regime as in *Figure 2*. This example shows the competition between three phage variants, all with $\phi_{\max} = 1$ but with varying response thresholds $\theta$. (B) Distribution of phage variants at evolutionary steady state for varying passaging episode durations $T$. In total 441 phage variants were included in this analysis, covering all combinations of $\phi_{\max}$ between 0 and 1 and $\theta$ between 0 and $cK$ with step sizes 0.05 and $0.05cK$, respectively. When the interval between passages is very short, again a fully lytic strategy ($\phi_{\max} = 0$) is selected. For longer times between passages, however, we consistently see that a strategy with $\phi_{\max} = 1$ and $\theta \approx 0.65cK$ dominates the population. The results shown for $T = 2$ h are representative for values of $T \leq 4$ h, while the results shown for $T = 12$ h represent results obtained for $T \geq 5$ h (see *Figure 3—figure supplement 1* for distributions for a large range of $T$-values). Almost identical results are obtained if instead of only phages a full sample (susceptible cells, lysogens, phages, and arbitrium) is passaged (*Figure 3—figure supplement 2*). (C) Rapid invasion by 'optimally' communicating phages into a population of phages with the 'optimal' bet-hedging strategy. The bet-hedging phages have $\phi = 0.04$ (see *Figure 2C*), while the communicating phages are characterised by $\phi_{\max} = 1$ and $\theta = 0.66cK$ (see panel C). The communicating phage is initialised at a frequency of 1% of the bet-hedging phage.

The online version of this article includes the following figure supplement(s) for figure 3:

**Figure supplement 1.** Phage variant distribution at evolutionary steady state for various lengths of the time interval between passages *T*.

**Figure supplement 2.** Phage variant distribution at evolutionary steady state for various lengths of the time interval between passages *T* under serial passaging of full samples (susceptible cells, lysogens, phages, and arbitrium).

(*Figure 4B*, green line). The same result was found for simulations of the competition between phage variants with different $\theta$-values under different effective burst sizes (*Figure 4B*, blue dots). We see that *Equation 6* provides a good prediction for the response threshold value that is selected over evolutionary time, especially for phages with high effective burst size (*Figure 4B*).

For phages with a very small effective burst size, the response threshold selected in the simulations tends to be lower than the analytical approximation. This is due to a violation of one of the simplifying assumptions made to arrive at the analytical approximation of *Equation 6*, namely that during the active epidemic the dynamics of the arbitrium concentration are dominated by its production through infections and arbitrium uptake and degradation by susceptible cells can be ignored. While this is a reasonable assumption in a fast progressing epidemic, it breaks down if the dynamics of the epidemic are slow, which is exactly the case if the effective burst size $bB$ is small. Under these conditions, the uptake and degradation of arbitrium by susceptible cells cause the arbitrium

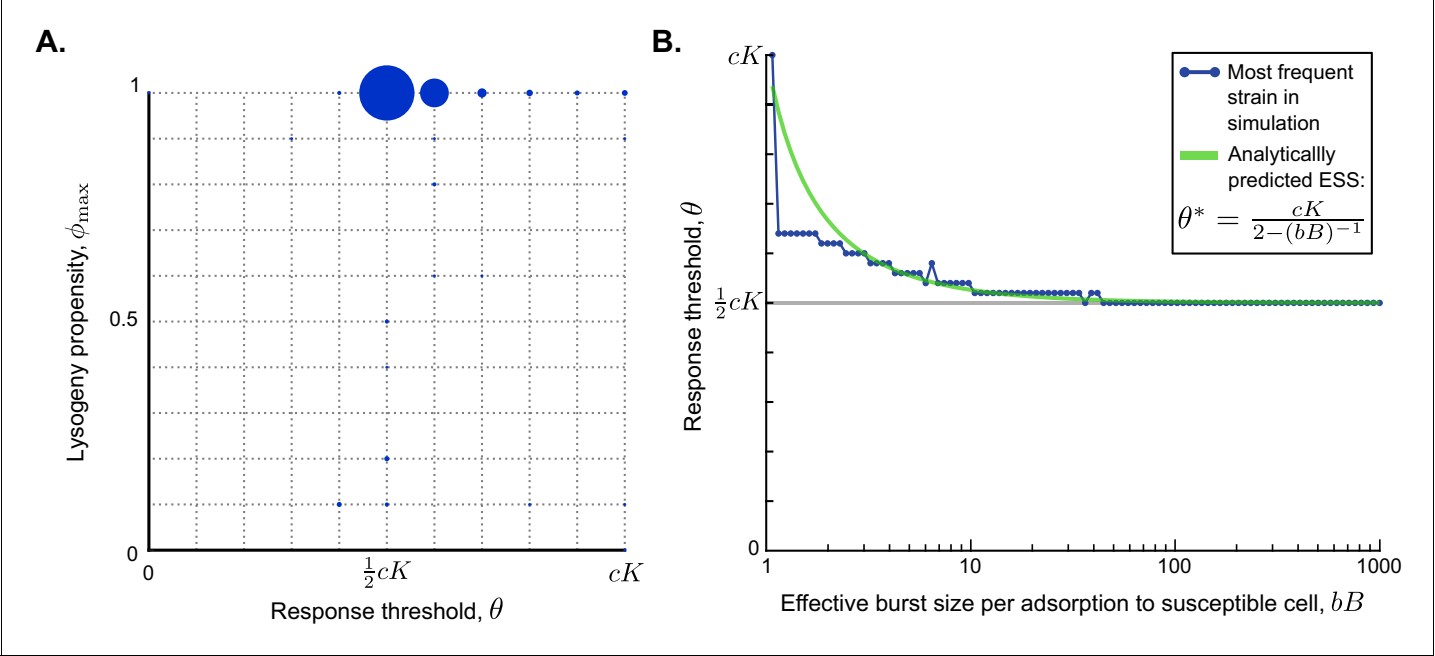

**Figure 4.** Parameter dependance of the selected values of $\phi_{max}$ and θ. (**A**) Parameter sweep results. A total of 500 simulations were run with randomly sampled parameters (**Table 1**) and a long time between passages ($T = 24$ h). In each simulation, 121 phage variants were included, covering all combinations of $\phi_{max} = 0$ to $\phi_{max} = 1$ and $\theta = 0$ to $\theta = cK$ with step sizes 0.1 and $0.1cK$, respectively. The size of the circles corresponds to the number of simulations that yielded that particular phage variant as most abundant at evolutionary steady state. (**B**) Analytically predicted θ-value as a function of the effective burst size per adsorption to a susceptible cell, $bB$, and most abundant phage variant found in a simulation with varying $bB$ but otherwise default parameter values, $T = 24$ h, $\phi_{max} = 1$ and $\theta = 0, 0.02cK, \ldots, cK$. The range on the x-axis is equal to the range sampled in the parameter sweep. The analytically derived evolutionarily stable $\theta^*$ is a good prediction for the response threshold selected in the simulations, especially for phages with high effective burst size.

concentration to be lower than assumed in the analytical derivation. Consequently, the actual selected response thresholds (which are essentially arbitrium concentration values) are lower than the analytically predicted values.

The result in *Equation 6* can be further understood biologically. Remember that the arbitrium concentration during the epidemic varies between $A = 0$ and $A = cK$, and is a reflection of the fraction of susceptible cells that have so far been infected. It makes sense that the evolutionarily stable response threshold causes phages to switch infection strategy somewhere in the middle of the epidemic: if a phage variant switches to the lysogenic strategy too early, its free phage population does not expand enough to compete with phages that switch later, but if it switches too late, the susceptible-cell density has decreased to such a degree that the phage has missed the window of opportunity for lysogen production. The ESS results from a balance between the fast production of phage progeny during the initial lytic cycles and the eventual production of sufficient lysogens. For phages with a high effective burst size, this balance occurs around the time that half of the available susceptible cells have been infected. Phages with lower effective burst size are, however, predicted to switch later, because these phages need to invest a larger portion of the available susceptible cells in the production of free phages to produce a sufficient pool of phages that can later form lysogens. Note, however, that the range of biologically reasonable effective burst sizes includes many high values (range of x-axis in *Figure 4B*, *Table 1*), that is, many real-life phages have high infectivity. Hence, for natural phages in general, we predict that if they evolve an arbitrium-like communication system, communication will be used to switch from causing mostly lytic to mostly lysogenic infections when in an outbreak approximately half of the pool of susceptible bacteria has been infected.

## Arbitrium communication is robust against variation in bacterial carrying capacity

So far, we have considered the evolution of arbitrium communication under highly predictable settings, with each outbreak taking place in a population of bacteria with the same initial density (i.e. the bacterial carrying capacity was constant). As argued above, in such a set-up the arbitrium concentration provides information on the density of susceptible cells still available for infection, and the phages use this to inform their lysis-lysogeny decision. While the bacterial carrying capacity can be kept constant in lab experiments, it is far from obvious that this would be the case in natural environments. This warrants the question of how robust the results are to variation in the bacterial carrying capacity.

We therefore performed simulations in which the carrying capacity varies from outbreak to outbreak. For a long time between passages (T = 24 h), at the start of each passaging episode a random carrying capacity was drawn from a gamma distribution with mean K and a pre-set variance that differs from simulation to simulation. We use the coefficient of variation (CV), which is defined as the standard deviation relative to the mean, to describe the level of noise.

*Figure 5* summarises the results of these additional simulations. Surprisingly, a communication strategy with $\phi_{\max} = 1$ and $\theta \approx 0.5cK$ is selected for a large range of carrying capacity noise up to CV $\leq 0.35$ (illustrated by CV = 0.22 in *Figure 5A*; see *Figure 5—figure supplement 1* for full data). In other words, even if the carrying capacity varies with a standard deviation up to one third of its mean value, the communication strategy described in the previous section is still selected.

As the coefficient of variation increases even further, the arbitrium response threshold value $\theta$ of the selected phages decreases, and so does the lysis-lysogeny propensity that is used at high arbitrium concentration $\phi_{\max}$ (*Figure 5B and C*). These results make sense: if the carrying capacity strongly varies between passaging episodes, the phages regularly cause outbreaks in bacterial populations with low density. Phages with a response threshold value larger than the bacterial carrying capacity do not produce any lysogens during such an outbreak, which is disastrous for their long-term fitness. Hence, lower response thresholds are selected. The corresponding lower $\phi_{\max}$ values likely evolve to compensate for the earlier switch to lysogen production caused by the lower $\theta$-

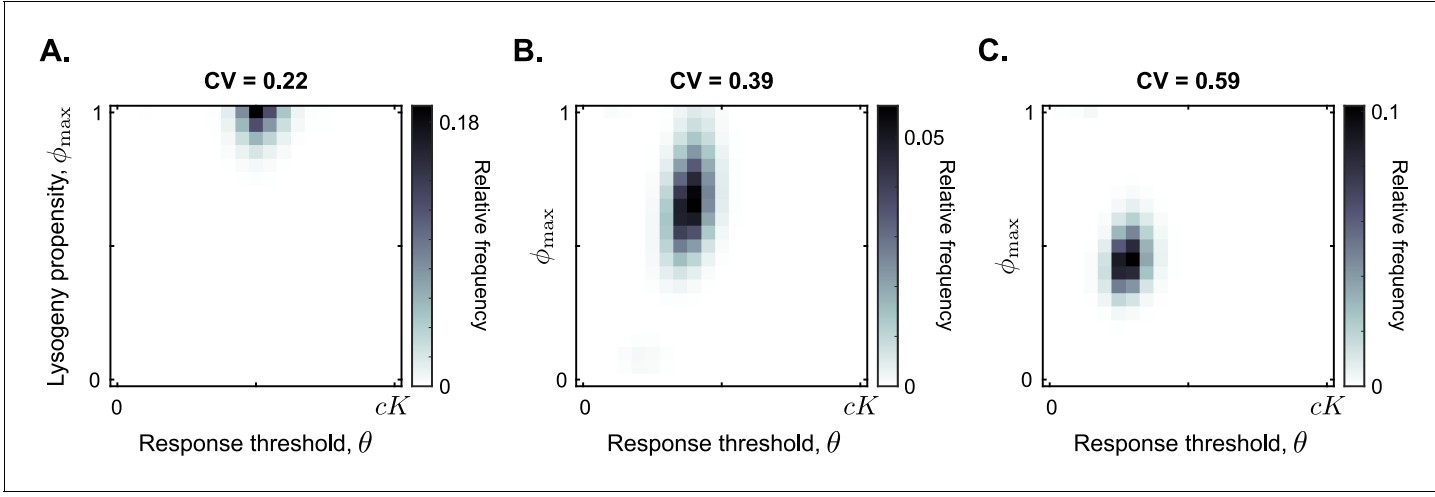

**Figure 5.** Distribution of phage variants at evolutionary steady state for increasing variation in the bacterial carrying capacity. Each simulation included a total of 441 phage variants that covered all combinations of $\phi_{\max}$ between 0 and 1 and $\theta$ between 0 and $cK$ with step sizes 0.05 and $0.05cK$, respectively. In each simulation, 1000 passaging episodes were simulated, with a long time between passages (T = 24 h). Parameter were set to default values (*Table 1*), except that at the start of each passaging episode the value of the bacterial carrying capacity was drawn from a gamma distribution with mean K. The coefficient of variation (CV = standard deviation / mean) was varied between simulations. Results are shown here for (A) CV = 0.22, (B) CV = 0.39, (C) CV = 0.59. The results in panel (A) are representative for CV $\leq 0.35$, and the results in panel (C) are representative for CV $\geq 0.5$ (*Figure 5—figure supplement 1*).

The online version of this article includes the following figure supplement(s) for figure 5:

**Figure supplement 1.** Phage variant distribution at evolutionary steady state for a sufficiently long time between passages (T = 24 h) and increasing variability in the bacterial carrying capacity between passaging episodes.

values. In highly variable conditions, phages are hence selected to switch from a lytic strategy very early in the epidemic to a bet-hedging strategy later.

While we find much lower response threshold values when the variation in bacterial carrying capacity is high, these threshold values do remain clearly larger than zero (*Figure 5C*). This is true even if the carrying capacity is exponentially distributed (CV = 1; see *Figure 5—figure supplement 1*). Hence, even under very high variation of bacterial density a form of arbitrium communication (in which phages use the arbitrium signal to switch from a lytic to a bet-hedging strategy) is still favoured over completely bet-hedging strategies.

## Discussion

We have presented a mathematical model of a population of phages that use an arbitrium-like communication system, and used this model to explore the evolution of the lysis-lysogeny decision and arbitrium communication under a serial-passaging regime. When arbitrium communication was excluded from the model, we found that bet-hedging phages with relatively low lysogeny propensity were selected. But when arbitrium communication was allowed to evolve these bet-hedging phages were outcompeted by communicating phages. These communicating phages switch from a lytic strategy early in the epidemic to a fully lysogenic strategy when approximately half of the available susceptible cells have been infected.

The serial-passaging set-up of the model is crucial for the evolution of the lysis-lysogeny decision and arbitrium communication. This has two main reasons. Firstly, it ensures that the phages are regularly exposed to susceptible cells, thus maintaining selection pressure on the lysis-lysogeny decision. Because of their high infectivity (see Materials and methods section and *De Paepe and Taddei, 2006*; *Wang, 2006*), most temperate phage outbreaks will completely deplete pools of susceptible bacteria, resulting in a bacterial population consisting of lysogens only in which the phage no longer replicates through infection (*Bossi et al., 2003*; *Gama et al., 2013*). The bet-hedging strategy we found in the absence of phage communication is a mechanism to deal with these (self-inflicted) periods of low susceptible cell availability, consistent with earlier studies (*Maslov and Sneppen, 2015*; *Sinha et al., 2017*). Secondly, the serial-passaging set-up imposes a dynamic of repeated epidemics in which a small number of phages is introduced into a relatively large pool of susceptible cells. Such dynamics are necessary for the arbitrium system to function: the arbitrium concentration provides a reliable cue for a phage's lysis-lysogeny decision only if it is low at the beginning of an epidemic and subsequently builds up to reflect the fraction of cells that have so far been infected.

Based on these considerations, we can stipulate which environments promote the evolution of small-molecule communication such as the arbitrium system. One major factor that can ensure a regular exposure to susceptible cells (the first requirement) is spatial structure. If phages mostly infect bacteria that are physically close to them, a global susceptible population can be maintained even though susceptible bacteria may be depleted in local environments (*Kerr et al., 2006*). Indeed, spatial structure has been shown to greatly influence phage evolution, for instance by promoting the selection of less virulent strains that deplete their local host populations more slowly (*Kerr et al., 2006*; *Heilmann et al., 2010*; *Berngruber et al., 2015*). For small-molecule communication to evolve, however, the phages would also have to undergo repeated, possibly localised, outbreak dynamics (the second requirement). Such dynamics could occur in structured meta-populations of isolated bacterial populations, between which the phages spread at a limited rate. Alternatively, phages might encounter large pools of newly susceptible bacteria if they escape superinfection immunity through mutation (*Zinder, 1958*; *Bailone and Devoret, 1978*; *Scott et al., 1978*). Under this scenario, however, any remaining arbitrium signal from previous infection events no longer provides accurate information about the number of susceptible cells available, since cells that were lysogenically infected are once again susceptible to infection with the new phage variant. If the escape mutation occurs after the arbitrium produced during previous epidemics has been degraded, this problem does not occur and the newly produced arbitrium does function as a reliable signal of susceptible cell density for the new phage variant. If, however, the escape mutation occurs while the arbitrium concentration is still high from previous outbreaks, the new phage variant will cause lysogenic infections while in fact the lytic cycle should be favoured. There will then be selection pressure on the new phage variant to acquire additional mutations that change its signal specificity. This

might in part explain the large diversity of phage signalling peptides observed (*Erez et al., 2017*; *Stokar-Avihail et al., 2019*).

The model presented in this paper allows us to put hypotheses about the arbitrium system to the test. For instance, it has been suggested that the arbitrium system would benefit from the production of arbitrium by lysogens, because phages thereby would be warned about the presence of neighbouring lysogens (which are immune to superinfection; *Hynes and Moineau, 2017*). Above we have argued, however, that under repeated epidemics, such as caused by serial passaging, selection on the lysis-lysogeny decision and arbitrium signalling is limited to the relatively short window of time in which all (locally) present susceptible cells become infected: afterwards no new infections occur and arbitrium therefore has no effect. During this short time window, the density of lysogens is still low, and any arbitrium produced by lysogens contributes little to the information already conveyed by arbitrium produced during infection events. Hence, our model predicts that, under repeated epidemics that completely deplete (local) pools of susceptible cells, the effects of arbitrium production by lysogens are likely very minimal. Arbitrium production by lysogens can be effective only if lysogens and susceptible cells coexist over sufficiently long periods of time, such that infection events occur in the presence of lysogens. In the model, we found that such coexistence is highly unlikely. Coexistence between lysogens and susceptible cells might, however, happen under circumstances that were not included in our model, for instance through a constant inflow of susceptible cells because of cell migration, or through the loss of superinfection immunity by lysogens.

Intriguingly, our model predicts that phages using small-molecule communication to coordinate their lysis-lysogeny decision would be selected to switch from a lytic to a lysogenic strategy once approximately half of the available susceptible bacteria have been lytically infected. This prediction warrants experimental testing. However, it also raises the question of how the phages would 'know' at what bacterial density the susceptible population has been halved. For the arbitrium signal to carry reliable information about the density of remaining susceptible cells, the initial concentration of susceptible bacteria has to be similar from outbreak to outbreak. Hence, one might expect the communication strategy to break down if the density of susceptible bacteria is variable. Surprisingly, this turned out not to be the case. We found that arbitrium-like communication could evolve even if the bacterial carrying capacity was highly variable. The characteristics of the communication system then depend on the level of noise. In highly variable environments, we predict the selection of phages that start their lysogen production earlier in an outbreak (i.e. phages that have a low response threshold), and then do so in a bet-hedging way (i.e. with a lysogeny propensity much smaller than 1).

In fact, few details are known so far about the response curve of phages' lysogeny propensity to the arbitrium concentration. In the model, we chose to implement the response to arbitrium as a stepwise function. This allowed us to clearly distinguish between strategies that are favoured at low arbitrium concentration (the lytic cycle) and at high arbitrium concentration (the lysogenic cycle). In reality, phages might respond more gradually to the arbitrium concentration. While this would alter some of our results (e.g. pinpointing an arbitrium concentration at which the phages switch infection strategy becomes harder, if not impossible), we do not expect the results in general to depend on the precise shape of the response curve: phages will still use the arbitrium signal to adjust their infection strategy to whichever strategy currently yields most progeny phage on the long run. Once more data become available on the actual shape of the response curve, these can be incorporated in the model by adjusting the arbitrium response function $\varphi(A)$, thus producing a more specific model of the arbitrium system.

Next to the arbitrium system, several other examples of temperate phages affected by small signalling molecules have recently been described. For instance, the *Vibrio cholerae*-infecting phage VP882 'eavesdrops' on a quorum-sensing signal produced by its host bacteria, favouring lytic over lysogenic infections when the host density is high (*Silpe and Bassler, 2019*), while in coliphages λ and T4 and several phages infecting *Enterococcus faecalis*, the induction of prophages, that is, the lysogeny-lysis decision, is affected by bacterial quorum sensing signals (*Ghosh et al., 2009*; *Rossmann et al., 2015*; *Laganenka et al., 2019*). The model could be adapted to capture these other regulation mechanisms by changing the arbitrium equation to an equation describing the production and degradation of the bacterial quorum sensing signal, and – for the second mechanism – letting the prophage reactivation rate $\alpha$, rather than the lysogeny propensity $\phi$, depend on the signal concentration. Similar analyses to the ones in this paper would then allow us to study under what

conditions phage eavesdropping on bacterial quorum sensing can and cannot evolve. Mathematical and computational modelling can thus help to better understand the ecology and evolution of these fascinating regulation mechanisms as well.

## Acknowledgements

We thank Rob J de Boer for valuable discussions and comments on the manuscript. This work was supported by the Human Frontier Science Program, grant nr. RGY0072/2015 (http://www.hfsp.org/funding/research-grants).

## Additional information

### Funding

| Funder | Grant reference number | Author |
|---|---|---|
| Human Frontier Science Program | RGY0072/2015 | Hilje M Doekes<br>Rutger Hermsen |

The funders had no role in study design, data collection and interpretation, or the decision to submit the work for publication.

### Author contributions

Hilje M Doekes, Conceptualization, Software, Formal analysis, Investigation, Visualization, Methodology, Writing - original draft, Writing - review and editing; Glenn A Mulder, Software, Investigation, Visualization, Writing - review and editing; Rutger Hermsen, Conceptualization, Formal analysis, Supervision, Funding acquisition, Methodology, Writing - review and editing

### Author ORCIDs

Hilje M Doekes (ID) https://orcid.org/0000-0002-6360-5176
Rutger Hermsen (ID) https://orcid.org/0000-0003-4633-4877

### Decision letter and Author response

Decision letter https://doi.org/10.7554/eLife.58410.sa1
Author response https://doi.org/10.7554/eLife.58410.sa2

## Additional files

### Supplementary files

• Transparent reporting form

### Data availability

All data were obtained through computer simulation. Scripts to run these simulations, simulated data, and analysis scripts are available at GitHub: https://github.com/hiljedoekes/PhageCom (copy archived at https://archive.softwareheritage.org/swh:1:rev:8124bcd18e18dd03b94a315b7694af3e-d2e4a002/).

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

## Appendix 1

### Model equations and parameters

#### A1.1 Full model equations, including mutations

For readability, the model equations in the main text (*Equation 1–4*) did not include mutations between phage variants. Here, we present the full model equations, including mutations. We assume that mutations happen when new phage particles (and hence new copies of the phage's genetic material) are formed prior to lysis of the host cell. Infection with a parent phage of variant $j$ results in progeny phages of variant $i$ with probability $\mu_{ij}$, where $\mu_{ii}$ is the probability that offspring of a parent phage of type-$i$ has no mutations. The full model reads:

$$\frac{\mathrm{d}S}{\mathrm{d}t} = rS(1 - N/K) - baSP, \tag{A.1}$$

$$\frac{\mathrm{d}L_i}{\mathrm{d}t} = rL_i(1 - N/K) + \varphi_i(A)baSP_i - \alpha L_i, \tag{A.2}$$

$$\frac{\mathrm{d}P_i}{\mathrm{d}t} = B\sum_j \mu_{ij}\big(\alpha L_j + [1 - \varphi_j(A)]baSP_j\big) - \delta P_i - aNP_i, \tag{A.3}$$

$$\frac{\mathrm{d}A}{\mathrm{d}t} = cbaSP - uNA, \tag{A.4}$$

with $\quad \varphi_i(A) = \phi_i \qquad\qquad$ if arbitrium communication is excluded from the model, $\qquad$ (A.5)

and $\quad \varphi_i(A) = \begin{cases} 0 & \text{if } A \le \theta_i, \\ \phi_{\max_i} & \text{if } A > \theta_i \end{cases}$ if arbitrium communication is included (see Figure 1B). $\quad$ (A.6)

In the first part of the manuscript, where we exclude arbitrium communication from the model, phage variants are characterised by their constant lysogeny propensity $\phi_i$ (*Equation A.5*). Mutations between phage variants were implemented in a stepwise fashion, that is, $\mu_{ij} = \mu > 0$ if $\phi_j$ is one step higher or lower than $\phi_i$, and $\mu_{ij} = 0$ $(i \neq j)$ otherwise. Throughout this study, a value of $\mu = 5 \cdot 10^{-4}$ was used. Varying $\mu$ alters the mutation-selection balance and thus affects the frequency of mutants in the quasi-species. As long as $\mu$ is reasonably small, however, it does not change which phage variant dominates the population, that is, $\mu$ does not affect the evolutionarily stable strategy.

In the second part of the manuscript, where we include the possibility of arbitrium communication, phage variants are characterised by two properties (Equation A.6): their arbitrium threshold $\theta_i$ and the lysogeny propensity obtained when the arbitrium concentration exceeds this threshold, $\phi_{\max_i}$. Mutations in the values of $\phi_{\max}$ and $\theta$ were implemented as independent processes, both happening in a stepwise fashion as explained above.

#### A1.2 Phage variants included in simulations

In the simulations of the restricted model (arbitrium communication excluded) shown in *Figure 2C*, a range of phage variants was included with $\phi_1 = 0$, $\phi_2 = 0.005$, ..., $\phi_{100} = 0.495$, $\phi_{101} = 0.5$. In the simulations presented in *Figure 2D*, a range of phage variants was included of $\phi_1 = 0$, $\phi_2 = 0.01$, ..., $\phi_{100} = 0.99$, $\phi_{101} = 1.0$.

In the simulations of the full model (arbitrium communication included) shown in *Figure 3B*, 441 phage variants were included, representing all possible combinations of $\phi_{\max} = 0, 0.05, \ldots, 1$ and $\theta = 0, 0.05cK, \ldots, cK$. In the simulations of *Figure 4A*, 121 phage variants were included representing all possible combinations of $\phi_{\max} = 0, 0.1, \ldots, 1$ and $\theta = 0, 0.1cK, \ldots, cK$. In the simulations of *Figure 4B*, all phage variants had $\phi_{\max} = 1$, but they varied in $\theta = 0, 0.02cK, \ldots, cK$.

All simulations were initialised with the susceptible cells at carrying capacity ($S = K$ cells per mL), no lysogens ($\sum_i L_i = 0$) and a low total density of $\sum_i P_i = 10^6$ phages per mL. All phage variants were initially present at equal frequency.

In all simulations mutations between phage variants were included as described in the previous section, with the exception of the competition experiment between the optimal bet-hedging phage variant and the optimal communicating phage variant (results shown in *Figure 3C*). In this latter case, the mutation rate was set to zero.

## A1.3 Parameter reduction

In total, the model of *Equation A.1–A.6* has nine parameters (ignoring mutation probabilities). This number can, however, be reduced by non-dimensionalising the equations. We introduce the dimensionless variables

$$\hat{t} = rt, \quad \hat{S} = \frac{S}{K}, \quad \hat{L}_i = \frac{L_i}{K}, \quad \hat{P}_i = \frac{P_i}{KB}, \quad \hat{A} = \frac{A}{cK}, \quad \hat{\theta}_i = \frac{\theta_i}{cK},$$

and define $\hat{P} = \sum_i \hat{P}_i$, $\hat{L} = \sum_i \hat{L}_i$, and $\hat{N} = \hat{S} + \hat{L}$. Let furthermore

$$\hat{B} = bB, \quad \hat{a} = \frac{aK}{r}, \quad \hat{\alpha} = \frac{\alpha}{r}, \quad \hat{\delta} = \frac{\delta}{r}, \quad \hat{u} = \frac{uK}{r}.$$

Using these new variables and parameters, the equations reduce to:

$$\frac{d\hat{S}}{d\hat{t}} = \hat{S}(1 - \hat{N}) - \hat{B}\hat{a}\hat{S}\hat{P}, \tag{A.7}$$

$$\frac{d\hat{L}_i}{d\hat{t}} = \hat{L}_i(1 - \hat{N}) + \hat{\varphi}_i(\hat{A})\hat{B}\hat{a}\hat{S}\hat{P}_i - \hat{\alpha}\hat{L}_i, \tag{A.8}$$

$$\frac{d\hat{P}_i}{d\hat{t}} = \sum_j \mu_{ij}\left(\hat{\alpha}\hat{L}_j + [1 - \hat{\varphi}_j(\hat{A})]\hat{B}\hat{a}\hat{S}\hat{P}_j\right) - \hat{\delta}\hat{P}_i - \hat{a}\hat{N}\hat{P}_i, \tag{A.9}$$

$$\frac{d\hat{A}}{d\hat{t}} = \hat{B}\hat{a}\hat{S}\hat{P} - \hat{u}\hat{N}\hat{A}, \tag{A.10}$$

with $\quad \hat{\varphi}_i(\hat{A}) = \phi_i \qquad\qquad$ if arbitrium communication is excluded from the model, $\tag{A.11}$

and $\quad \varphi_i(A) = \begin{cases} 0 & \text{if } A \leq \hat{\theta}_i, \\ \phi_{\max_i} & \text{if } A > \hat{\theta}_i \end{cases} \quad$ if arbitrium communication is included. $\tag{A.12}$

Five dimensionless parameters are left in *Equation A.7–A.12*: the effective burst size per adsorption of a phage to a susceptible cell, $\hat{B}$, and the scaled rates $\hat{a}$, $\hat{\alpha}$, $\hat{\delta}$, and $\hat{u}$. These non-dimensionalised equations are used throughout the rest of this appendix, unless stated otherwise, dropping the hats for convenience.

## Appendix 2

### Equilibrium analysis

In this section, we find the dynamical equilibria existing in the model, and derive parameter conditions for their stability. This analysis provides us with a baseline expectation of the densities of phages, lysogens, and susceptible cells that the model converges to after sufficient time.

Equilibria of the model are found by equating *Equation A.7–A.10* to zero and solving for the model variables. By the definition of equilibrium, the arbitrium concentration at equilibrium is constant:

$$\bar{A} = \begin{cases} \dfrac{Ba\bar{S}\bar{P}}{u\bar{N}} & \text{if } \bar{N}>0, \\ \bar{A}=0 & \text{if } \bar{N}=0, \end{cases} \tag{A.13}$$

where the bar indicates equilibrium values. Because the equilibrium arbitrium concentration is constant, at equilibrium the different phage variants are characterised by their lysogeny propensity $\varphi_i(\bar{A})$ only, irrespective of whether arbitrium communication is included in the model or not. Hence, expressions for the model equilibria are the same in the absence and presence of arbitrium communication. Below, we derive these expressions, and determine under what conditions the different model equilibria are stable.

In the model, four qualitatively different types of equilibria can occur.

Firstly, there is a trivial equilibrium at $\bar{S}=0$, and $\bar{L}_i=0$, $\bar{P}_i=0$ for all phage variants $i$. This equilibrium is unstable as long as the bacterial logistic growth rate satisfies $r>0$.

Secondly, there is an equilibrium in which the susceptible population is at carrying capacity and the infection is absent: $\bar{S}=1$, and $\bar{L}_i=0$, $\bar{P}_i=0$ for all phage variants $i$. This equilibrium is stable if no phage-lysogen pairs $P_i$-$L_i$ can invade on the susceptible population. To derive stability conditions, we approximate the dynamics of $P_i$ and $L_i$ in the vicinity of the equilibrium by the linearised equations

$$\begin{pmatrix} \frac{dL_i}{dt} \\ \frac{dP_i}{dt} \end{pmatrix} \approx \begin{pmatrix} -\alpha & \bar{\phi}_i Ba \\ \alpha & (1-\bar{\phi}_i)Ba-\delta-a \end{pmatrix} \begin{pmatrix} L_i \\ P_i \end{pmatrix} =: J \begin{pmatrix} L_i \\ P_i \end{pmatrix}, \tag{A.14}$$

where $\bar{\phi}_i = \varphi_i(0)$ is the lysogeny propensity of phage type $i$ at the equilibrium. No phage-lysogen pair can invade (i.e. the equilibrium is stable) precisely if the real parts of both eigenvalues of the Jacobian matrix $J$ are negative for all $i$. The eigenvalues of the Jacobian matrix $J$ are given by

$$\lambda_{+/-} = \frac{\Gamma \pm \sqrt{\Delta}}{2}, \quad \text{with } \Gamma = a\big(B(1-\bar{\phi}_i)-1\big)-\delta-\alpha \text{ and } \Delta = \Gamma^2 + 4\alpha(a(B-1)-\delta).$$

The real parts of $\lambda_{+/-}$ are both negative precisely if $\Delta<\Gamma^2$ and $\Gamma<0$. From $\Gamma<0$, we find

$$a(B(1-\bar{\phi}_i)-1)<\delta+\alpha, \tag{A.15}$$

while $\Delta<\Gamma^2$ yields

$$a(B-1)<\delta. \tag{A.16}$$

Since all parameters are non-negative and $0 \leq \bar{\phi}_i \leq 1$, the condition in *Equation A.16* is more stringent than the condition in *Equation A.15*. Note also that the condition in *Equation A.16* does not depend on the lysogeny propensity of the invading phages, $\bar{\phi}$. Hence, the equilibrium $\bar{S}=1$, $\bar{L}_i=0$, and $\bar{P}_i=0$ for all phage variants $i$ is stable exactly if the condition in *Equation A.16* is satisfied. This condition makes sense: phages cannot spread in a susceptible cell population at carrying capacity if their infection rate $Ba\bar{S}$ is smaller than the decay rate of phage particles $\delta+a\bar{S}$.

Thirdly, there is a class of equilibria in which $\bar{S}=0$, and $\bar{L}_i>0$, $\bar{P}_i>0$ for some $i$. In these equilibria, the total densities of lysogens $\bar{L}$ and of free phages $\bar{P}$ are given by

$$\bar{L} = 1-\alpha, \quad \bar{P} = \frac{\alpha\bar{L}}{\delta+a\bar{L}} = \frac{\alpha(1-\alpha)}{\delta+a(1-\alpha)}. \tag{A.17}$$

In the absence of susceptible cells at equilibrium, no infections can take place and hence all phage variants behave identically (since phages vary only in $\varphi_i(A)$, which occurs exclusively in the infection terms). This is reflected in the equations for the different lysogen variants, which for $\bar{S} = 0$ and $\bar{L} = 1 - \alpha$ (*Equation A.17*) reduce to

$$\frac{dL_i}{dt} = L_i(1 - \bar{L}) - \alpha L_i = 0.$$

Hence, any combination of $\bar{L}_i$ values with $\sum_i \bar{L}_i = \bar{L} = 1 - \alpha$ and corresponding $\bar{P}_i$-values,

$$\bar{P}_i = \frac{\alpha \sum_j \mu_{ij} \bar{L}_j}{\delta + a(1-\alpha)}, \tag{A.18}$$

is an equilibrium. Analogous to the reasoning above, such an equilibrium is stable if the susceptible cells cannot invade the phage-lysogen population at equilibrium. The linearised equation for the dynamics of $S$ near the equilibrium of *Equation A.17* reads

$$\frac{dS}{dt} \approx S(1 - \bar{L}) - BaS\bar{P} = S\left(\alpha - \frac{Ba\alpha(1-\alpha)}{\delta + a(1-\alpha)}\right). \tag{A.19}$$

The right-hand side of this equation is negative (i.e. susceptible cells cannot invade) precisely if $\delta + a(1-\alpha) < Ba(1-\alpha)$, or summarised

$$\delta < a(B-1)(1-\alpha). \tag{A.20}$$

(Note that to arrive at condition *Equation A.20* we assume $\delta + a(1-\alpha) > 0$. This assumption is justified because the spontaneous induction rate of lysogens $\alpha$ is small, and hence $1 - \alpha > 0$ (see *Table 1*).)

Lastly, there can be an equilibrium in which the susceptible cells, lysogens and phages all coexist. Expressions for $\bar{S}$, $\bar{P}_i$, and $\bar{L}_i$ at this equilibrium are bulky and not directly insightful. This type of equilibrium was however extensively analysed in recent work by Wahl et al. for phages with a constant lysogeny propensity (i.e. the restricted model where arbitrium communication is excluded) (*Wahl et al., 2018*). Remember that in an equilibrium state, phage variants are characterised by their lysogeny propensity $\varphi_i(\bar{A})$ only (i.e. differences in response threshold $\theta$ are relevant only if they are reflected in differences in $\varphi_i(\bar{A})$; phage variants $i$ and $j$ with $\theta_i \neq \theta_j$ but $\varphi_i(\bar{A}) = \varphi_j(\bar{A})$ can for all practical purposes be considered the same), and hence the results found by Wahl et al. can be extended to the model analysed here. Phage variants with different lysogeny propensity $\varphi_i(\bar{A})$ can be seen as consumers that compete for a single resource, namely susceptible cells to infect. As long as $\bar{S} > 0$, we hence expect competitive exclusion, and the phage population at equilibrium will be dominated by phages with a single lysogeny propensity value $\bar{\phi} \equiv \varphi(\bar{A})$ (when mutations are ignored, these phages will be the only ones present; otherwise we find a quasispecies). Furthermore, Wahl et al. show that if susceptible host cells coexist with a resident lysogen-phage population with some lysogeny propensity $\bar{\phi}_r$, a phage-lysogen pair of a variant with higher lysogeny propensity $\bar{\phi}_i > \bar{\phi}_r$ can always invade on this equilibrium. Hence, in this equilibrium, the dominant phage will be the one with the highest equilibrium lysogeny propensity, $\bar{\phi} = 1$.

As has been demonstrated previously (*Stewart and Levin, 1984*; *Wahl et al., 2018*), the 'coexistence equilibrium' is stable only if phages and lysogens can invade on a susceptible population at carrying capacity (i.e. the condition in *Equation A.16* is violated), and susceptible cells can invade on the phage-lysogen population in equilibrium (i.e. the condition in *Equation A.20* is violated). Hence, susceptible cells, lysogens and phages all coexist precisely if

$$(1-\alpha)(B-1)a < \delta < (B-1)a. \tag{A.21}$$

If the effective burst size $B > 1$ (a necessary condition for the phage to be viable), this can be rewritten as

$$(1-\alpha) < \frac{\delta}{(B-1)a} < 1. \tag{A.22}$$

Because the spontaneous induction rate of lysogens, $\alpha$, is small ($\alpha < 0.01$, see *Table 1*), the condition in *Equation A.22* is very specific. Susceptible cells, lysogens and phages coexist only if the exponential growth rate of a lytic phage spreading in a susceptible population at carrying capacity, $(B-1)a - \delta$, is positive but very small, that is, if the epidemic is viable but only barely so. In reality, however, most phage epidemics are characterised by a high infectivity, mainly because of a large burst size (*De Paepe and Taddei, 2006*). Therefore, the condition in *Equation A.22* is rarely satisfied, and for most phages we should instead expect to converge to equilibria of the third type ($\bar{S} = 0, \bar{L} > 0, \bar{P} > 0$).

This observation has consequences for the selection pressures on phage variants over the course of a typical epidemic. As soon as the pool of susceptible host cells is depleted, competition between the different phage variants vanishes and the relative frequency of the variants freezes (see *Figure 2—figure supplement 1* for an illustration). Under these conditions, no infections take place and hence there is no selection on the lysis-lysogeny decision. We conclude that the evolution of the lysis-lysogeny decision of typical phages requires regular perturbations away from equilibrium conditions.

## Appendix 3

## Derivation of the evolutionarily stable strategy (ESS) under the serial-passaging regime

In the main text, we present simulation results of the evolution of the lysis-lysogeny decision of phages exposed to a serial-passaging regime, both when arbitrium communication is excluded from the model (*Figure 2*), and when it is included (*Figures 3* and *4*). These simulations show that, after many passaging episodes, a single variant dominates the phage population (accompanied by its quasispecies, see *Figure 2C*, *Figure 3B*). We therefore assume that, within the parameter range of interest, a pure Evolutionarily Stable Strategy (ESS) exists. In this section, we derive analytical expressions for the ESS. We first provide a definition of the ESS under a serial-passaging regime, and give a general description of how this ESS can be found (section A3.1). Then, we apply this general approach to derive the ESS of phages that differ in a constant lysogeny propensity, $\phi$ (absence of arbitrium communication, section A3.2), and the ESS of communicating phages that differ in their response threshold $\theta$ (section A3.3).

### A3.1 General approach

Consider a population of phages under a serial-passaging regime with long time $T$ between passages. At the start of each passaging episode, a fraction $D$ of the phages is taken from the end of the previous episode, and is added to a 'fresh' population of susceptible bacteria at carrying capacity. This procedure is repeated over many episodes. Within each episode, the dynamics of the susceptible bacteria, lysogens, phages, and arbitrium are described by *Equation A.7–A.12*.

An evolutionarily stable strategy (ESS) is defined as a strategy that cannot be invaded by any other strategy that is initially rare. To find the ESS, we therefore consider a scenario where an invader phage variant attempts to invade a resident phage variant. Below, we specify what it means for a phage variant to be able to invade in a resident phage population under the imposed serial-passaging regime.

Envision a resident population consisting of an isogenic phage population that has gone through many passaging episodes. Over time, the dynamics within these episodes have converged to a repeatable trajectory characterised by $P_r(t)$, the resident phage density over time, $S(t)$, the density of susceptible bacteria over time, and $L_r(t)$, the density of lysogens over time. At the start of one episode, now suddenly introduce a second phage with its own (possibly different) strategy. Crucially, the initial density of this invading phage $P_i(0)$ is infinitesimally small. Consequently, during the first passaging episode the dynamics of the resident phage and the bacteria ($P_r(t)$, $S(t)$, and $L_r(t)$) are not affected by the invader. The invader is able to invade precisely if at the end of the first episode its frequency has increased relative to that of the resident, that is, if $P_i(T)/P_r(T)>P_i(0)/P_r(0)$.

Note that the relative frequency of an invader with exactly the same strategy as the resident does not change during an episode. Suppose that such an invader is the best-performing invader under the environment set up by the resident; then this implies that no invader can increase in frequency over a passaging episode, and therefore the resident strategy must be an ESS. In other words, in a resident phage population consisting of the ESS only, the ESS itself is the optimal strategy for an invading phage variant, that is, *the ESS is the optimal response to itself*.

What does it take to be the best-performing invader? To answer this question, we consider the dynamics within a single passaging episode in more detail.

If the time between passages $T$ is long (see below for exact condition), and the parameter conditions are such that the system converges to an equilibrium with $S=0, P>0, L>0$ (typical parameter values, see Appendix 2), the dynamics within an episode can be divided in three distinct phases (see *Figure 2—figure supplement 1*):

1. *Epidemic phase*. A substantial population of susceptible bacteria ($S>0$) supports an ongoing epidemic. Free phages and lysogens are formed through infection of susceptible bacteria.
2. *Transition phase*. The population of susceptible cells has collapsed ($S\approx0$). The lysogen population expands to fill up the space left behind by lysed cells. Free phages particles can no longer infect susceptible cells, and disappear through decay and adsorption to lysogens.
3. *Equilibrium phase*. The composition of the population is well-characterised by an equilibrium of 'type 3' (see *Equation A.17*). There is a small but consistent population of free phages that

originates from lysogens through spontaneous induction. The distribution of phage variants in the free phage population is a direct representation of the relative frequency of the variants in the lysogen population (*Equation A.18*).

Let $T_E$ be the time at which the susceptible population collapses, that is, the end of the epidemic phase. If the time between passages $T$ is sufficiently larger than $T_E$, the passage takes place during the equilibrium phase. The relative frequency of phage variants in the passaged sample then directly reflects the relative frequency of the corresponding lysogens. Since lysogens are only differentially formed through infection dynamics (and not through lysogen replication, which happens at the same rate for all lysogen variants), the relative frequency of the different lysogens is established during the epidemic phase and does not change afterwards. The performance of an invading phage can hence be measured by its lysogen production between $t = 0$ and $t = T_E$.

The dynamics of $S(t)$, and consequently $T_E$, are determined by the resident phage: highly virulent resident phages (that cause many lytic infections, for instance because of a low $\phi$-value) deplete the susceptible cell population faster than less virulent residents. The optimal invader under a certain resident is the phage variant that produces the most lysogens during the limited window of opportunity that it is offered by the environment set up by the resident. Since the ESS is the optimal response to itself, it is the strategy that, as an invader, produces the most lysogens during an epidemic phase set up by itself. We will use this reasoning to identify the ESS.

## A3.2 Evolutionarily stable lysogeny propensity of non-communicating phages

Consider the restricted model in which arbitrium communication is excluded and phages are characterised by a fixed lysogeny propensity $\phi$. To find the ESS under this scenario, we take the following steps:

1. Derive how the duration of the epidemic, $T_E$, depends on the $\phi$-value of a resident phage population.
2. Find the optimal $\phi$ given a fixed value of $T_E$, that is, the $\phi$-value that yields a maximal lysogen density at time $T_E$.
3. Combine 1. and 2. to find the ESS: the $\phi$-value $\phi^*$ that maximises its lysogen density at time $T_E(\phi^*)$, the duration of the epidemic as dictated by its own $\phi$-value, $\phi^*$.

This approach is summarised in *Box 1*. Below, we provide the full analysis.

### A3.2.1 Simplifying assumptions

To make the model analytically tractable, we make the following simplifying assumptions (based on the typical infection dynamics, see *Figure 2A*):

1. Bacterial growth, decay of free phages, and induction of lysogens are considered to be much slower than the phage infection dynamics. We hence ignore these processes when describing the epidemic phase.
2. The epidemic ends when all susceptible cells have been infected. In other words, we solve $T_E$ from $\int_{t=0}^{T_E} BaSP_r \, \mathrm{d}t = 1$ (where this one represents the carrying capacity in the non-dimensionalised units).
3. The density of lysogens during the epidemic is small, hence $N \approx S$.
4. The susceptible population remains relatively constant for some time, after which it rapidly collapses. We approximate these dynamics with a block function, setting $S = 1$ for $t \leq T_E$, and $S = 0$ for $t > T_E$.

Under these assumptions, the dynamics of the resident phage population for the period $0 \leq t \leq T_E$ are described by:

$$\frac{\mathrm{d}P_r}{\mathrm{d}t} = (Ba(1 - \phi_r)S - aS)P_r = (B(1 - \phi_r) - 1)aP_r, \tag{A.23}$$

with solution

$$P_r(t) = P_{r,0}e^{(B(1-\phi_r)-1)at} = P_{r,0}e^{(\eta - \phi_r)Bat}, \tag{A.24}$$

where $P_{\mathrm{r},0} \equiv P_{\mathrm{r}}(0)$ is the initial density of resident phages and we have introduced $\eta := 1 - B^{-1}$. Note that the description of the infection dynamics in *Equation A.23* is meaningful only if early in the epidemic the phage population indeed grows exponentially, that is, if $\phi_{\mathrm{r}} < \eta$. For default parameter settings, this upper bound on $\phi_{\mathrm{r}}$ is well above the $\phi$-values that are typically selected ($\phi \approx 0.04$ [*Figure 2C*], while $\eta = 0.5$ [*Table 1*]), indicating that this assumption is reasonable.

## A3.2.2 Duration of the epidemic given a resident phage

First, we derive how the duration of the epidemic, $T_{\mathrm{E}}$, depends on the lysogeny propensity of a resident phage variant $\phi_{\mathrm{r}}$. To find $T_{\mathrm{E}}(\phi_{\mathrm{r}})$, we substitute *Equation A.24* into assumption 2:

$$\int_{t=0}^{T_{\mathrm{E}}} BaSP_{\mathrm{r}}(t)\,\mathrm{d}t = \int_{t=0}^{T_{\mathrm{E}}} BaP_{\mathrm{r},0}\mathrm{e}^{(\eta-\phi_{\mathrm{r}})Bat}\,\mathrm{d}t = \frac{P_{\mathrm{r},0}}{(\eta-\phi_{\mathrm{r}})}\left(\mathrm{e}^{(\eta-\phi_{\mathrm{r}})BaT_{\mathrm{E}}}-1\right),$$

and then equate this integral to one to find

$$T_{\mathrm{E}}(\phi_{\mathrm{r}}) = \frac{1}{Ba(\eta-\phi_{\mathrm{r}})}\log\left(\frac{\eta-\phi_{\mathrm{r}}}{P_{\mathrm{r},0}}+1\right). \tag{A.25}$$

Note that the density of the resident phage at time $T_{\mathrm{E}}$ is now given by

$$P_{\mathrm{r}}(T_{\mathrm{E}}) = P_{\mathrm{r},0}\mathrm{e}^{(\eta-\phi_{\mathrm{r}})BaT_{\mathrm{E}}} = P_{\mathrm{r},0}+\eta-\phi_{\mathrm{r}}. \tag{A.26}$$

Therefore, the expression for $T_{\mathrm{E}}$ (*Equation A.25*) can also be read as:

$$T_{\mathrm{E}}(\phi_{\mathrm{r}}) = \frac{1}{Ba(\eta-\phi_{\mathrm{r}})}\log\left(\frac{P_{\mathrm{r}}(T_{\mathrm{E}})}{P_{\mathrm{r},0}}\right). \tag{A.27}$$

*Equation A.27* will prove useful later for the derivation of the ESS.

## A3.2.3 Optimal invader strategy given a fixed duration of the epidemic

Next, we ask what invader lysogen propensity $\phi_{\mathrm{i},\mathrm{opt}}$ maximises the invader's lysogen production, $L_{\mathrm{i}}(T_{\mathrm{E}})$, if the duration of the epidemic $T_{\mathrm{E}}$ is fixed. For $0 \leq t \leq T_{\mathrm{E}}$, the dynamics of $L_i(t)$ are described by

$$\frac{\mathrm{d}L_{\mathrm{i}}}{\mathrm{d}t} = \phi_{\mathrm{i}}BaP_{\mathrm{i}}. \tag{A.28}$$

Since $P_{\mathrm{i}}(t) = P_{\mathrm{i},0}\mathrm{e}^{(\eta-\phi_{\mathrm{i}})Bat}$ (see *Equation A.24*) and $L_{\mathrm{i}}(0) = 0$, we can now solve

$$L_{\mathrm{i}}(t) = P_{\mathrm{i},0}\frac{\phi_{\mathrm{i}}}{\eta-\phi_{\mathrm{i}}}\left(\mathrm{e}^{(\eta-\phi_{\mathrm{i}})Bat}-1\right). \tag{A.29}$$

To find the $\phi_{\mathrm{i}}$-value that maximises $L_{\mathrm{i}}(T_{\mathrm{E}})$, we take the derivative of *Equation A.29* with respect to $\phi_{\mathrm{i}}$:

$$\frac{\partial L_{\mathrm{i}}(T_{\mathrm{E}})}{\partial \phi_{\mathrm{i}}} = P_{\mathrm{i},0}\left(\frac{\eta(\mathrm{e}^{(\eta-\phi_{\mathrm{i}})BaT_{\mathrm{E}}}-1)}{(\eta-\phi_{\mathrm{i}})^2}-\frac{\phi_{\mathrm{i}}BaT_{\mathrm{E}}\mathrm{e}^{(\eta-\phi_{\mathrm{i}})BaT_{\mathrm{E}}}}{\eta-\phi_{\mathrm{i}}}\right). \tag{A.30}$$

To find $\phi_{\mathrm{i},\mathrm{opt}}$, we should hence solve

$$\frac{P_{\mathrm{i},0}}{\eta-\phi_{\mathrm{i}}}\left(\frac{\eta(\mathrm{e}^{(\eta-\phi_{\mathrm{i}})BaT_{\mathrm{E}}}-1)}{\eta-\phi_{\mathrm{i}}}-\phi_{\mathrm{i}}BaT_{\mathrm{E}}\mathrm{e}^{(\eta-\phi_{\mathrm{i}})BaT_{\mathrm{E}}}\right) = 0. \tag{A.31}$$

Unfortunately, *Equation A.31* cannot be solved analytically. We can however simplify *Equation A.31* by noting that for sufficiently small $\phi_{\mathrm{i}}$, $(\eta-\phi_{\mathrm{i}})$ is of order $0.1-1$, while $T_{\mathrm{E}}$ is typically of order 1, and $Ba$ is of order $10-1000$ (*Table 1*). Hence, $\mathrm{e}^{(\eta-\phi_{\mathrm{i}})BaT_{\mathrm{E}}}$ is typically $\gg 1$, and *Equation A.31* can be approximated by

$$\frac{P_{i,0}e^{(\eta-\phi_i)BaT_E}}{\eta-\phi_i}\left(\frac{\eta}{\eta-\phi_i}-Ba\phi_iT_E\right)=0. \tag{A.32}$$

From *Equation A.32* we find

$$
\begin{aligned}
&\frac{\eta}{\eta-\phi_i}=Ba\phi_iT_E\\
\iff\quad & BaT_E\phi^2-\eta BaT_E\phi_i+\eta=0\\
\iff\quad & \phi_{i,opt}(T_E)=\tfrac{1}{2}\eta\pm\tfrac{1}{2}\sqrt{\eta^2-\frac{4\eta}{BaT_E}}.
\end{aligned}\tag{A.33}
$$

### A3.2.4 The ESS

*Equation A.25* and its alternative formulation *Equation A.27* describe how the duration of the epidemic $T_E$ depends on the lysogeny propensity $\phi_r$ of the current resident, while *Equation A.33* gives the value $\phi_{i,opt}$ that maximises the lysogen production of an invader during an epidemic of a fixed duration $T_E$. The ESS is now given by the value $\phi^*$ that is 'optimal' as defined by *Equation A.33*, when it itself is the resident and hence dictates $T_E(\phi^*)$. Combining *Equation A.33* and *Equation A.27* we find

$$
\begin{aligned}
&\phi^*=\tfrac{1}{2}\eta\pm\tfrac{1}{2}\sqrt{\eta^2-4\frac{\eta(\eta-\phi^*)}{\log(P_r(T_E)/P_{r,0})}}\\
\iff\quad & (\phi^*-\tfrac{1}{2}\eta)^2=\tfrac{1}{4}\eta^2-\frac{\eta(\eta-\phi^*)}{\log(P_r(T_E)/P_{r,0})}\\
\iff\quad & (\phi^*)^2-\phi^*\left(\eta+\frac{\eta}{\log(P(T_E)/P_0)}\right)+\frac{\eta^2}{\log(P_r(T_E)/P_{r,0})}=0,
\end{aligned}
$$

from which we can solve:

$$\phi^*=\frac{1}{2}\eta\left(1+\frac{1}{\log(P_r(T_E)/P_{r,0})}\right)\pm\frac{1}{2}\eta\left(1-\frac{1}{\log(P_r(T_E)/P_{r,0})}\right). \tag{A.34}$$

Of these two solutions, $\phi_+=\eta$ is an asymptote at which our approximation no longer holds (remember that we previously demanded that $\phi_r<\eta$ to ensure initial spread of the infection). Hence, $\phi^*$ should be given by $\phi_-$:

$$\phi^*=\frac{\eta}{\log(P_r(T_E)/P_{r,0})}. \tag{A.35}$$

Although *Equation A.35* seems to provide an elegant equation for the ESS, it still depends on $P_r(T_E)$ and $P_{r,0}$. If the interval between passages is sufficiently long, the phage density at the end of a passaging episode will be given by *Equation A.17* and hence

$$P_{r,0}=D\frac{\alpha(1-\alpha)}{\delta+a(1-\alpha)}, \tag{A.36}$$

where $D$ is the dilution factor of phages upon passaging. While the value of $P_{r,0}$ does not depend on the lysogeny propensity $\phi_r$, $P_r(T_E)$ does (see *Equation A.26*). Substituting $P_r(T_E)=P_{r,0}+\eta-\phi^*$ yields

$$\phi^*=\frac{\eta}{\log(1+\frac{\eta-\phi^*}{P_{r,0}})}. \tag{A.37}$$

This equation cannot be solved analytically. However, we can make a reasonable approximation of *Equation A.37* by considering the differences in orders of magnitude of the terms within the logarithm. As argued above, $(\eta-\phi^*)$ is generally of order $0.1-1$, while typical values of $P_{r,0}$ are several orders of magnitude smaller ($P_{r,0}\approx10^{-5}$). Therefore, we can approximate the logarithm in *Equation A.37* by

$$\log(1 + \frac{\eta - \phi^*}{P_{r,0}}) \approx \log\left(\frac{1}{P_{r,0}}\right).$$

Using this approximation, we find

$$\phi^* = \frac{\eta}{\log(\frac{1}{P_{r,0}})}, \tag{A.38}$$

which is also presented in the main text (*Equation 5*). *Equation A.38* and *Equation A.36* were used to find the analytical predictions shown in *Figure 2D*.

## A3.3 Evolutionarily stable response threshold of communicating phages

Next, consider a population of phages that do engage in arbitrium communication, again under a serial-passaging regime with sufficiently long time between the passages. Below, we use an approach similar to section A3.2, but more general, to derive the evolutionarily stable arbitrium response threshold, $\theta^*$. We take the following steps:

1. Describe the dynamics of an invading phage and its corresponding lysogens in an environment dictated by a resident phage.
2. Find the optimal invader response threshold under a fixed resident response threshold, that is, find the θ-value that maximises the invader's lysogen production at time $T_E$ when the dynamics of susceptible cells (and hence $T_E$) are determined by a fixed resident phage.
3. Determine the ESS, $\theta^*$, as the optimal response to itself: the optimal invader response threshold (as found in step 2) if that same response threshold is the resident strategy.

We found that the results below are best understood in terms of the non-scaled model; in particular the (non-scaled) burst size of the phages turns out to be an important parameter. Therefore, the derivations below are presented for the dimensionalised equations *Equation A.1–A.6*.

### A3.3.1 Simplifying assumptions

To make the model tractable, we again make a few simplifying assumptions:

1. As in section A3.2, we assume that there is a separation of time scales between the infection dynamics of the phages and the reproduction of the bacteria, spontaneous phage decay and lysogen induction. Hence, when describing the epidemic phase we ignore these other processes.
2. Additionally, we assume that there also is a separation of time scales between the production of arbitrium through infections (first term in *Equation A.4*) and its uptake and degradation by cells (second term in *Equation A.4*). We ignore the uptake and degradation of arbitrium during the early epidemic, such that the increasing arbitrium concentration reflects the decrease of the susceptible cell density because of infections.
3. We assume that communicating phages switch from a completely lytic strategy ($\varphi(A) = 0$) to a completely lysogenic strategy ($\varphi(A) = 1$) once the arbitrium concentration exceeds the phages' response threshold. This is in line with observations from simulations, where we find that phage variants with $\phi_{max} = 1$ dominate the population for a wide range of parameter values (*Figure 4A*).

The assumptions above are less strict then the assumptions made in section A3.2. In particular, we no longer assume that the density of susceptible cells, $S(t)$, remains constant for the duration of the epidemic $0 \le t \le T_E$. Rather, for the derivations below it suffices to assume that $S(t)$ is a declining function which is completely determined by the resident phage, and that $S(t)$ is sufficiently close to 0 after the epidemic, that is, for times $t > T_E$ (where $T_E$ still depends on the characteristics of the resident phage).

It will be useful to refer to the *time* when the resident and invader switch to lysogeny as $\tau_r$ and $\tau_i$, respectively. Given particular dynamics of the susceptible cell density and the arbitrium concentration dictated by a resident phage population, there is a direct relation between $\tau_i$ and $\theta_i$ (the arbitrium response threshold of the invader). Keep in mind, however, that this relation changes if the resident phage is changed.

## A3.3.2 Dynamics of the invading phage and its corresponding lysogens

First, we describe how the dynamics of an invading phage variant depend on the resident phage population. Remember that we consider an invader phage variant that starts off at infinitesimally small density, and attempts to invade an isogenic resident phage population that has already converged to a repeatable trajectory of $P_r(t)$, $L_r(t)$, $S(t)$, and $A(t)$ per passaging episode. Under these conditions, the dynamics of $S(t)$, $N(t) = S(t) + L_r(t)$ and $A(t)$ over the first passaging episode do not depend on the switch time $\tau_i$ of the invader, but only on the switch time of the resident phage, $\tau_r$. Based on the assumptions formulated above, the ODEs for the density of the invading phage and its corresponding lysogens can be written as

$$\frac{dP_i(t|\tau_i, \tau_r)}{dt} = \begin{cases} BbaS(t|\tau_r)P_i(t|\tau_i, \tau_r) - aN(t|\tau_r)P_i(t|\tau_i, \tau_r), & (t < \tau_i) \\ -aN(t|\tau_r)P_i(t|\tau_i, \tau_r), & (t \geq \tau_i) \end{cases} \quad \text{(A.39)}$$

$$\frac{dL_i(t|\tau_i, \tau_r)}{dt} = \begin{cases} 0, & (t < \tau_i) \\ baS(t|\tau_r)P_i(t|\tau_i, \tau_r), & (t \geq \tau_i) \end{cases} \quad \text{(A.40)}$$

where vertical lines are used to indicate which phage charactics the trajectories of variables depend upon. *Equation A.39–A.40* capture the switch from a completely lytic infection strategy (for $t \leq \tau_i$), in which new phage particles are produced through infection but no lysogens are formed, to a completely lysogenic strategy (for $t > \tau_i$), in which no new phage particles are produced but all infections result in the production of lysogens (see assumption 3). Remember that we here use the dimensionalised equations, so $B$ is the burst size, $a$ the rate of adsorption of phages to bacterial cells (irrespective of whether they are susceptible or lysogen), and $b$ the probability that adsorption to a susceptible cell leads to an infection.

The solution to *Equation A.39* can be written as

$$P_i(t|\tau_i, \tau_r) = P_{i,0} \times \begin{cases} \exp\left(Bba \int_0^t S(t'|\tau_r)dt' - a \int_0^t N(t'|\tau_r)dt'\right), & (t < \tau_i) \\ \exp\left(Bba \int_0^{\tau_i} S(t'|\tau_r)dt' - a \int_0^t N(t'|\tau_r)dt'\right), & (t \geq \tau_i) \end{cases} \quad \text{(A.41)}$$

as is easily verified by differentiating this solution with respect to $t$.

As before, the performance of the invading phage variant is determined by its lysogen production during the epidemic phase, that is, between $t = 0$ and $t = T_E$. At any time $t$, the density of invader lysogens is

$$L_i(t|\tau_i, \tau_r) = \begin{cases} 0, & (t < \tau_i) \\ \int_{\tau_i}^t baS(t'|\tau_r)P_i(t|\tau_i, \tau_r)dt'. & (t \geq \tau_i) \end{cases} \quad \text{(A.42)}$$

Invading phage variants are selected on their lysogen density at the end of the epidemic, $L_i(T_E|\tau_i, \tau_r)$. Once the epidemic phase has ended ($t \geq T_E$), no new phage particles or lysogens are formed through infection. Hence, any reasonable switching time must obey $\tau_i < T_E$. Furthermore, since $S(t) \approx 0$ for any time $t \geq T_E$,

$$L_i(T_E|\tau_i, \tau_r) = \int_{\tau_i}^{T_E} baS(t'|\tau_r)P_i(t'|\tau_i, \tau_r)dt' = \int_{\tau_i}^{\infty} baS(t'|\tau_r)P_i(t'|\tau_i, \tau_r)dt' \quad \text{(A.43)}$$

which we will denote $\Lambda_i(\tau_i, \tau_r)$.

## A3.3.3 Optimal invader strategy given some resident phage

The optimal invader strategy $\tau_i$ given $S(t|\tau_r)$ and $N(t|\tau_r)$ is the one that maximises $\Lambda_i(\tau_i, \tau_r)$. To find this optimal strategy, we differentiate *Equation A.43* with respect to $\tau_i$:

$$\frac{\partial \Lambda_i(\tau_i, \tau_r)}{\partial \tau_i} = -baS(\tau_i|\tau_r)P_i(\tau_i|\tau_i, \tau_r) + \int_{\tau_i}^{\infty} baS(t'|\tau_r)\frac{\partial P_i(t'|\tau_i, \tau_r)}{\partial \tau_i}dt'. \quad \text{(A.44)}$$

The derivative in the integrand can be calculated from *Equation A.41* (noting that, inside the integral, $t' \geq \tau_i$):

$$\frac{\partial P_i(t|\tau_i, \tau_r)}{\partial \tau_i} = \frac{\partial}{\partial \tau_i}\left(Bba \int_0^{\tau_i} S(t'|\tau_r)dt'\right)P_{i,0} \exp\left(Bba \int_0^{\tau_i} S(t'|\tau_r)dt' - a \int_0^t N(t'|\tau_r)dt'\right)$$

$$= BbaS(\tau_i|\tau_r)P_i(t|\tau_i, \tau_r).$$

(A.45)

Inserting the last expression into *Equation A.44* yields

$$\frac{\partial \Lambda_i(\tau_i, \tau_r)}{\partial \tau_i} = -baS(\tau_i|\tau_r)P_i(\tau_i|\tau_i, \tau_r) + Bb^2a^2S(\tau_i|\tau_r)\int_{\tau_i}^{\infty} S(t'|\tau_r)P_i(t'|\tau_i, \tau_r)dt'$$

$$= -baS(\tau_i|\tau_r)P_i(\tau_i|\tau_i, \tau_r) + BbaS(\tau_i|\tau_r)\Lambda_i(\tau_i, \tau_r).$$

(A.46)

The terms in *Equation A.46* have a clear interpretation. By taking the derivative of $\Lambda_i(\tau_i, \tau_r)$ to $\tau_i$, we are implicitly comparing one possible invading phage variant (phage 1) that switches at time $t = \tau_i$ to a second invading phage variant (phage 2) that switches ever so slightly later, at $t = \tau_i + d\tau$. *Equation A.46* says that the lysogen density of these two variants at the end of the epidemic will differ because of two effects: On the one hand (first term) phage 2 will have a *lower* lysogen density than phage 1 because it does not produce lysogens in the time interval from $\tau_i$ to $\tau_i + d\tau$. The damage is $-baS(\tau_i|\tau_r)P_i(\tau_i|\tau_i, \tau_r)d\tau$ lysogens per volume. On the other hand, phage 2 will have a higher *higher* lysogen density because it produces additional free phages in the time interval from $\tau$ to $\tau + d\tau$, which results in additional lysogens in the rest of the epidemic. As a result, throughout the rest of the epidemic the second phage has $(1 + BbaS(\tau_i|\tau_r)d\tau)$ times as many phages as the first phage variant, and therefore produces an additional number of $BbaS(\tau_i|\tau_r)L_i(\tau_i, \tau_r)d\tau$ lysogens per volume.

The optimal invading phage variant given a resident phage is the variant with the value of $\tau_{i,\text{opt}}(\tau_r)$ for which the two terms in *Equation A.46* cancel precisely:

$$P_i(\tau_{i,\text{opt}}(\tau_r)|\tau_{i,\text{opt}}(\tau_r), \tau_r) = B\Lambda_i(\tau_{i,\text{opt}}(\tau_r), \tau_r).$$

(A.47)

That is, the optimal invader switches precisely when its phage density is equal to its total eventual lysogen production multiplied by the burst size *B*.

We may rewrite *Equation A.47* as

$$BE_i(\tau_{i,\text{opt}}(\tau_r)|\tau_r) = 1,$$

(A.48)

where $E_i(\tau_i|\tau_r) \equiv \Lambda_i(\tau_i, \tau_r)/P_i(\tau_i|\tau_i, \tau_r)$ is the number of lysogens eventually produced per phage of the invader phage variant present at time $\tau_i$. $E_i(\tau_i|\tau_r)$ can be interpreted as a kind of 'exchange rate', expressing the value of a single phage at time $\tau_i$ in the currency of lysogens. This suggests another way of phrasing the results above, where we compared two phage variants of which phage 2 switched slightly later than phage 1: During the time interval from $\tau_{i,1}$ to $\tau_{i,2} = \tau_{i,1} + d\tau$, both competing invading phage variants infect $baS(\tau_{i,1})P_i(\tau_{i,1}|\tau_{i,1}, \tau_r)d\tau$ susceptible bacteria per volume. Phage 1 directly converts these infected bacteria into lysogens. Phage 2 instead converts each of them into *B* additional phages. Whether this is a good idea depends precisely on whether increasing the phage density by *B* phages per volume will, during the rest of the epidemic, result in an increased lysogen density of more than 1 lysogen per volume. That is, phage 2 is the better invader precisely if $BE_i(\tau_i|\tau_r) > 1$, while phage 1 is the better invader if $BE_i(\tau_i|\tau_r) < 1$. Again we see that the optimal invader must obey *Equation A.47* and *Equation A.48*.

## A3.3.4 The ESS

To find the ESS, we ask what phage variant is the optimal response to itself, that is, what phage variant satisfies

$$\tau_{i,\text{opt}}(\tau^*) = \tau^*.$$

(A.49)

In other words, the ESS must obey a special case of *Equation A.47* and *Equation A.48*:

$$P_i(\tau^*|\tau^*, \tau^*) = B\Lambda_i(\tau^*, \tau^*) \quad \text{or} \quad BE_i(\tau^*|\tau^*) = 1.$$

(A.50)

Importantly, in this case the resident and invader behave identically, so that the exchange rate of the resident must be the same as that of the invader. That is,

$$E_i(\tau^*|\tau^*) = E_r(\tau^*|\tau^*) = \Lambda_r(\tau^*)/P_r(\tau^*|\tau^*), \tag{A.51}$$

where $\Lambda_r(\tau_r)$ is the total density of lysogens eventually produced by a resident with switching time $\tau_r$. Combined with *Equation A.50* this results in:

$$B\Lambda_r(\tau^*)/P_r(\tau^*|\tau^*) = 1. \tag{A.52}$$

Hence, the ESS is the strategy that, when it is the only phage variant present, switches from the lytic to the lysogenic cycle precisely when the density of free phage particles it has is equal to the burst size times the density of lysogens it will still produce in the remainder of the active epidemic (see explanation in the previous section).

So far, we have expressed the results for the ESS as a switching time $\tau^*$. In reality, however, the communicating phages switch when a certain threshold arbitrium concentration $\theta^*$ is reached. As a last step, we therefore have to relate the terms in *Equation A.52* to the arbitrium concentration. Under our simplifying assumptions, the arbitrium dynamics between time $t=0$ and $t=\tau_r$ are described by

$$A(t|\tau_r) = cbaS(t|\tau_r)P_r(t|\tau_r), \tag{A.53}$$

where $c$ is increase in arbitrium concentration per infection. The total arbitrium concentration at time $t$ is hence given by

$$A(t|\tau_r) = \int_0^t cbaS(t'|\tau_r)P_r(t'|\tau_r)dt', \tag{A.54}$$

which can be written as $A(t|\tau_r) = cI_r(t|\tau_r)$, where

$$I_r(t|\tau_r) = \int_0^t baS(t'|\tau_r)P_r(t'|\tau_r)dt' \tag{A.55}$$

is the infection density: the number of infections that has occurred per volume at time $t$.

To express the ESS in terms of the arbitrium concentration, we first show that $P_r(\tau^*|\tau^*)$ is approximately proportional to $I_r(\tau^*|\tau^*)$. In general, the resident phage density obeys an equation equivalent to *Equation A.39* (even though this equation was originally written down for the invading phage). For the time period $t<\tau_r$, the solution of this equation can be expressed as

$$\begin{aligned} P_r(t|\tau_r) &= P_{r,0} + \int_0^t (Bb-1)aS(t'|\tau_r)P_r(t'|\tau_r)dt' \\ &= P_{r,0} + (B-b^{-1})I_r(t|\tau_r) \end{aligned} \tag{A.56}$$

Provided that the initial phage density $P_{r,0}$ is negligible compared to the phage density at time $\tau^*$, we find that

$$P_r(\tau^*|\tau^*) \approx (B-b^{-1})I_r(\tau^*|\tau^*). \tag{A.57}$$

Next, we use that the epidemic will eventually consume (almost) all susceptible bacteria. (Note that this is equivalent with our earlier assumption that $S(t)\approx 0$ for $t>T_E$.) Hence, we must have that

$$\Lambda_r(\tau^*) \approx S(0) - I_r(\tau^*|\tau^*). \tag{A.58}$$

If we insert *Equation A.57* and *Equation A.58* into *Equation A.47* and solve for $I_r(\tau^*|\tau^*)$, we arrive at

$$I_r(\tau^*|\tau^*) = \frac{S(0)}{2-(bB)^{-1}}. \tag{A.59}$$

That is, the ESS switches when the infection density obeys *Equation A.59*. This implies that the ESS should have the threshold

$$\theta^* = cI_{\mathrm{r}}(\tau^*|\tau^*) = \frac{cK}{2 - (bB)^{-1}}, \tag{A.60}$$

where we have substituted $S(0) = K$, the carrying capacity of the bacteria. *Equation A.60* is also presented in the main text (*Equation 6*). This equation was used to provide the analytical estimates shown in *Figure 4B*.

