## [Decision Letter]

**Acceptance summary:**

The recent discovery of phage communication allowing collective decision making (Erez et al., 2017), raised many questions about the social lives of viruses. This paper uses a mathematical modeling approach to determine the conditions under which a small molecule communication system among phages governing the lysis-lysogen decision would evolve. The paper, the first to employ theory on the topic, demonstrates the evolutionary advantages of phage communication over a bet-hedging strategy, which was previously believed to govern life cycle decisions in viruses.

**Decision letter after peer review:**

Thank you for submitting your article "Repeated outbreaks drive the evolution of bacteriophage communication" for consideration by *eLife*. Your article has been reviewed by three peer reviewers, including Samuel L Díaz-Muñoz as the Reviewing Editor and Reviewer #1, and the evaluation has been overseen by Aleksandra Walczak as the Senior Editor.

The reviewers have discussed the reviews with one another and the Reviewing Editor has drafted this decision to help you prepare a revised submission.

Summary:

This paper uses a modeling approach to determine the conditions under which a small peptide-mediated communication as a strategy informing lysis-lysogeny decision would arise in viruses. This topic arises from the recent discovery of phage communication by Erez et al., 2017. It starts using traditional differential equation system to investigate the lysis-lysogeny decision in a bet hedging context. However, most of the manuscript goes on to describe a modeling strategy mimicking serial passaging to examine the bet hedging strategy and the communication strategies, including examining how they work in a head to head competition (when does the communication strategy invade). Finally, the paper examines the threshold of proportion of cells infected at which phages are predicted to switch to lysogeny, coming up with 0.5 as a consistent number across a wide parameter swath. The manuscript demonstrates how such a communication mechanism can be fitter than a bet-hedging strategy where some fixed fraction of infections result in lysogeny.

In sum, this manuscript is very clear, well-argued, well-written, and it was a pleasure to read it. This paper is a brilliant contribution not only for being (surprisingly) the first to address this topic from a theoretical perspective, but for its thoughtfulness, rigor, completeness, and the connection of the theoretical and the biological. There are a few areas for improvement, but this manuscript should be an outstanding contribution, certainly to the emergent sociovirology literature, but also to a wide audience including microbiologists, virologists, ecologists, evolutionary biologists, and behaviorists.

Please pay very special attention to the “Revisions expected in follow-up work” section below in revising your manuscript, the other revisions are there for our consideration and are given with the intention of improving the manuscript.

1) Some assumptions of the model should be better justified and/or discussed, for the reader to assess the generality of the conclusions obtained in the manuscript. In particular:

a) Subsection “Model” paragraph three: It would be good to comment on how realistic the stepwise assumption is for phi(A).

b) Results third paragraph: This discussion strongly relies on the fact that cells stop dividing as carrying capacity is reached. In reality, cells may die at a certain rate, or some cells may leave the system or be lost, implying that some rate of division could be sustained at equilibrium. It would be really good to discuss this possibility and the impact it may have on results. The serial passaging scenario considered afterwards is a special way of including such an effect, but at discrete and periodic times.

Fourth paragraph: It would be important to further motivate the serial passaging scenario that is chosen. Is it related to actual or potentially realistic experiments? Are the conclusions obtained robust to modifications of this scenario?

2) Results second paragraph: I completely agree with the statements and rationale outlined in this paragraph. However, the scenario described in paragraph four should then be part of the Materials and methods. I still think it can be included in the Results as is, but needs to be included in the Materials and methods.

3) Some elements of Discussion should be specified:

(3a) Paragraph four: Two mutations occurring in a very short lapse of time seems rather unlikely a priori. Could the authors comment on whether scenarios where one happens and then the other would be favorable? Related to mutations: In the passaging is there selection with regard to the phage mutations or are they likewise proportionally represented in the next passage? If they are rare in the system, they could be lost in the dilution. How would this affect the potential evolution and impact the model outcomes?

b) Could there be other scenarios under which the production of arbitrium by lysogens is more useful?

c) Final paragraph: It would be great if the authors could give some indications about how the current model could be adapted to these other cases and what insight it may bring. This would add value to the manuscript by potentially making its scope broader.

4) The readability of Figure 1A, which I think needs revision. I can see the authors likely spent considerable effort on it, but it remains very difficult to parse. Is it really necessary to have four different kinds of arrows? As the eye wanders around the diagram, it's not clear where to start or where to end. Perhaps it would help to organizing cycles more neatly into circles? The many virus pictograms add visual clutter without aiding clarity. I don't have a clear recipe, but I'd encourage the authors to solicit input from colleagues outside their field so as to make this figure maximally accessible, especially for a broad-audience-journal like *eLife*. Elsewhere the authors do a truly admirable job, reminding the reader what their parameters are etc., so it would be a shame for the reader to stumble at Figure 1A.

5) The discussion is brief, but excellent. It really covers a lot of the questions that I wanted to know before reading the paper and does an amazing job of connecting model results to biology and making predictions to test empirically.

Revisions expected in follow-up work:

1) My major "science concern" was spurred right from the start by the sentence in the Abstract, "our model predicts the selection of phages that switch infection strategy when half of the available susceptible cells have been infected". Seeing as it is notoriously difficult for a virus to estimate what fraction of cells were infected, this raised a worry that persisted through reading the entire paper – until the very last paragraph, where it was finally discussed. My suggestions would be to do this (a) sooner, and (b) better.

a) Sooner: I'd recommend explicitly commenting on the issue already in the Introduction and at least promise that it will be discussed.

b) Better: “The arbitrium concentration during the early epidemic then is a direct reflection of the fraction of susceptible cells that have so far been infected”

The readout is a direct reflection only of how many cells were lysed. In the model, yes, this is the same as fraction. But as the authors acknowledge at the very end this requires the assumption that bacterial cells were at carrying capacity K. In natural conditions, the carrying capacity can differ due to environmental factors – "unknown" to the virus.

Far from being a minor worry, the issue is directly relevant for the authors' comparison between the communicating and the bet-hedging strategy. Under the communicating strategy, if the carrying capacity were modulated to e.g. half of its usual value, it seems that the virus would never switch to the lysogenic phase – a crippling blow to fitness. In contrast, the bet-hedging strategy seems vastly more robust to such a modulation. This suggests that the conclusions could be changed quite strongly if the carrying capacity were picked randomly (within some range) between passaging trials.

It sounds like there is a prediction here.

I do not intend this comment as negative / undermining authors' findings. Quite the opposite, I think by adding this analysis the authors could strengthen their argument, demonstrating the utility of a quantitative framework like theirs, and perhaps even make additional predictions. Beyond what level of (passage-to-passage) variability of K the bet-hedging strategy becomes favored? Addressing this comprehensively would be a project in itself; but generating a figure panel illustrating this in simulations would strengthen the authors' case for the benefits of modeling work.

An optional point:

Since the cell-mediated uptake and degradation of arbitrium is also mediated by bacterial cells, and is thus also dependent on bacterial cell density, it's possible that this might mitigate the problem (making the arbitrium concentration less strongly dependent on K than naïve linearity). I don't know if it's true, but if it is, it would be cool, and worth highlighting. Unless I'm mistaken, this is not currently discussed. I think that could strengthen the story.

2) Figure 3: In the competition between bet-hedging and communicating variants, what is the impact of initial conditions? Moreover, here, only the 2 top variants of each type are competed against each other. Is this sufficient to conclude that communicating variants will always be selected by competition? If binary comparisons are sufficient, this should be stated and explained. If not, then it would be important to show explicitly what happens when various phages of both types compete together.

3) Scripts really should be openly available unless there is a compelling reason not to do so, especially in this case where the results depend so closely on the code. I would strongly urge the authors to instead publish all simulation scripts alongside the manuscript, both for archiving purposes but also to facilitate access.

4) The parameters tested should be in the main text. Note that, other than the construction of the equations according to life history parameters, this is the only connection to "biology" that non-theoretical researchers will have to the manuscript. Table A1 should be in the main text. The table should list the conventional units of the parameters. For instance does burst size of 2 refer to 2 virions? This value would normally be in the 10's^-1^00's for most phages. Some of these parameters don't seem reasonable at first glance. For instance, the adsorption rate is usually something on the order of 10^-10^ to 10^-12^. I suspect this could be because these are "scaled" values. If the scaled values are made for model convenience they should also be listed separately. This way the connection between the model and the biology will be more evident to readers.

5) The idea of having to go away from a steady state for this to work is very appealing. It is also likely much closer to biological reality in this case. Other than models that specifically address serial passaging, is the approach of disturbing ESS's regularly a new concept? Is there a generalizable framework for this?

6) Some conclusions should be specified or clarified:

a) Figure 2 B-C: Some diversity seems to survive at steady state for T>5h. I believe this is what the authors refer to as "quasi-species" in the manuscript. It would be important to discuss and interpret this surviving diversity, and to explain whether it is going to survive forever or to slowly decay, and why.

b) Figure 4: Please explain the difference between the analytical prediction and the simulation results at small values of bB in Figure 4B. In particular, does any of the simplifying assumptions result in this discrepancy?

---

## [Author Response]

Revisions for this paper:1) Some assumptions of the model should be better justified and/or discussed, for the reader to assess the generality of the conclusions obtained in the manuscript. In particular:a) Subsection “Model” paragraph three: It would be good to comment on how realistic the stepwise assumption is for phi(A).

To our knowledge, few details are currently known about the response function *φ*(A). Therefore, we had to make some simplifying assumption about the shape of this curve. We do agree that it is fair to acknowledge this, and to discuss this assumption. We now do so in the Discussion.

b) Results third paragraph: This discussion strongly relies on the fact that cells stop dividing as carrying capacity is reached. In reality, cells may die at a certain rate, or some cells may leave the system or be lost, implying that some rate of division could be sustained at equilibrium. It would be really good to discuss this possibility and the impact it may have on results. The serial passaging scenario considered afterwards is a special way of including such an effect, but at discrete and periodic times.

We think a misunderstanding has arisen here. The susceptible host cell population is depleted at equilibrium not because cells no longer divide, but because the susceptible host cells are outcompeted by lysogens that are immune to superinfection and hence do not experience the additional death rate induced by phage infections. We have altered the text to stress that the susceptible cells are outcompeted by lysogens.

The logistic terms used to describe bacterial growth in our model do not assume that cells stop dividing when carrying capacity is reached. Rather, they assume that at carrying capacity the division rate of cells is equal to the death (or loss) rate, such that a stable density of bacteria is maintained. In other words: the logistic growth rate is a net growth rate (division – death).

To clarify this point, let us show mathematically that an additional density-independent death rate can always be absorbed into a logistic term of net growth. Assume that the bacterial division rate is density dependent (and is hence described by a logistic-like term), while the bacterial death rate does not depend on density. The net growth of bacterial density *B* is then described by dBdt=dB(1−bh)−dB where *b* is the bacterial division (or birth) rate, *h* is the bacterial density at which the division rate is equal to zero, and *d* is the density-independent death rate. Now, note that we can rewrite this equation as: dBdt=(b−d)B−bB2h=(b−d)B (1−b(b−d)hB) If we now define *r = b/d* and *K = (1 – d/b)h*, this equation is exactly equal to the logistic terms used in our model.

Fourth paragraph: It would be important to further motivate the serial passaging scenario that is chosen. Is it related to actual or potentially realistic experiments? Are the conclusions obtained robust to modifications of this scenario?

On the one hand, the serial passaging scenario was inspired by the set-up of actual serial passaging experiments (e.g., Bull et al., Evolution, 1993; Bollback et al., Mol Biol and Evol, 2007). On the other hand, it is also meant to mimic the large changes in the density of available susceptible host cells probably experienced by phages in real life situations. This is now stated more clearly in the Materials and methods.

In terms of robustness, several modifications to the serial passaging set-up can be imagined. We chose to study two: varying the carrying capacity of bacteria between cycles, and altering which entities are passaged. The first of these modifications was also motivated by another comment (“Revisions expected in follow-up work – 1b”) and is discussed in detail there. In summary, we found the results to be quite robust against these variations in bacterial carrying capacity (see new Figure 5 and subsection “Arbitrium communication is robust against variation in bacterial carrying capacity”). For the second modification, note that in our initial set-up only phages are transferred from one epidemic cycle to the next. To test the robustness of the results against alterations of this assumption, we repeated our simulations but now passaged a small sample of the full system (*i.e.*, susceptible bacteria, lysogens, phages, and arbitrium). We found that the results were highly robust to this change in the set-up. For the case without arbitrium signalling these results are shown in the new (Figure 2—figure supplement 2 ), and for the case with arbitrium signalling they are shown in the new Figure 3—figure supplement 2.

2) Results second paragraph: I completely agree with the statements and rationale outlined in this paragraph. However, the scenario described in paragraph four should then be part of the Materials and methods. I still think it can be included in the Results as is, but needs to be included in the Materials and methods.

We understand this concern. We have moved the technical description of the serial passaging regime to the Materials and methods, while maintaining our explanation of why such a regime is necessary in the Results.

3) Some elements of Discussion should be specified:(3a) Paragraph four: Two mutations occurring in a very short lapse of time seems rather unlikely a priori. Could the authors comment on whether scenarios where one happens and then the other would be favorable?

Indeed, it is not really necessary for these two mutations to take place within a very short lapse of time. A mutation that allows phages to escape superinfection immunity will always free up a new pool of susceptible cells, namely the lysogens that are now no longer protected against this altered phage. If this mutant phage variant however still responds to the arbitrium that was produced during previous infections of the original phage variant and this arbitrium is still present in a sufficiently high concentration, the phage likely makes the “wrong” lysis-lysogeny decision: to cause lysogenic infections and hence replicate slowly while it could replicate much more quickly through the lytic cycle. Under these circumstances, there will be selection pressure on the mutant phages to acquire additional mutations that alter the signalling peptide.

We have altered the discussion to describe this scenario.

Related to mutations: In the passaging is there selection with regard to the phage mutations or are they likewise proportionally represented in the next passage?

There is no selection during the passaging and phage variants are proportionally included in the passaged sample. This is described in the introduction of the serial passaging set-up (Materials and methods).

If they are rare in the system, they could be lost in the dilution. How would this affect the potential evolution and impact the model outcomes?

Since our model is a deterministic model of continuous densities and not a stochastic model of discrete individuals, loss by dilution is impossible. The density of particular strains may become very low but is never actually zero. This is a consequence of the choice of modelling framework, and obviously not a true representation of nature. If a certain phage variant is very rare in a natural system, it can of course be lost by dilution. However, since we consider phage variants that can be formed through mutations of other phages, such a loss could always be compensated by a mutation that reinstates the lost variant. How quickly such a mutant arises depends on population characteristics such as the effective population size. Demographic stochasticity (including loss by dilution) can hence quantitatively alter the evolutionary dynamics. However, as long as the dynamics are not completely dominated by such stochasticity (*i.e.*, selection is not completely overshadowed by drift), the qualitative results such as the ESS will be the same.

b) Could there be other scenarios under which the production of arbitrium by lysogens is more useful?

This is a fair question, and we have given it considerable thought. The main reason that the production of arbitrium by lysogens is not beneficial in our modelling set-up is that lysogens and susceptible cells do not coexist for any significant amount of time. If there are no susceptible cells in the presence of lysogens, then there are no infection events and hence arbitrium produced by the lysogens cannot influence any lysis-lysogeny decisions. As we show in the analysis of the model’s equilibria and the parameter conditions for these equilibria to be stable, coexistence between susceptible cells and lysogens in equilibrium is very unlikely. However, coexistence could be attained under different circumstances. The most reasonable scenario is then that somehow there is a constant influx of susceptible cells into a population of lysogens.

We see two ways in which this could happen. Firstly, there might be an actual, physical inflow of susceptible cells. In natural systems, large-scale outbreaks could then correspond to phage colonization of a new area (*e.g.,* a particular part of a human gut), while the low-rate constant inflow is caused by movement of cells in this area (*e.g.*, bacterial migration within the gut).

Secondly, a low constant influx of susceptible cells into a population of lysogens could arise from the occurrence of loss-of-super immunity mutations in the prophages carried by the lysogens which render the lysogen vulnerable to the phage again. Since such mutations likely happen at low rates the “influx rate” of susceptible cells will be low, but such a process might allow a small population of susceptible cells to coexist with a population of lysogens. Since there will be only few of these susceptible cells at any given time, a lysogenic strategy is favoured when these cells are infected, and phages might hence benefit from arbitrium production by lysogens.

We now discuss both possibilities in the Discussion.

c) Final paragraph: It would be great if the authors could give some indications about how the current model could be adapted to these other cases and what insight it may bring. This would add value to the manuscript by potentially making its scope broader.

Thank you for this suggestion. We have added a short description of possible adaptations to our model and what we might learn from them.

4) The readability of Figure 1A, which I think needs revision. I can see the authors likely spent considerable effort on it, but it remains very difficult to parse. Is it really necessary to have four different kinds of arrows? As the eye wanders around the diagram, it's not clear where to start or where to end. Perhaps it would help to organizing cycles more neatly into circles? The many virus pictograms add visual clutter without aiding clarity. I don't have a clear recipe, but I'd encourage the authors to solicit input from colleagues outside their field so as to make this figure maximally accessible, especially for a broad-audience-journal like eLife. Elsewhere the authors do a truly admirable job, reminding the reader what their parameters are etc., so it would be a shame for the reader to stumble at Figure 1A.

We understand this comment and have tried to make the figure easier to digest. We removed many virus pictograms and restructured the diagram such that the figure can be read starting in the upper left corner. The infection event and subsequent lysis-lysogeny decision now take center stage. The lytic and lysogenic cycle are shown as two equally shaped ellipses, to stress the fact that these are two alternative life cycles. We have furthermore included a colour correspondence between the process influenced by the arbitrium concentration (the lysis-lysogeny decision) and the arbitrium itself.

Revisions expected in follow-up work:1) My major "science concern" was spurred right from the start by the sentence in the Abstract, "our model predicts the selection of phages that switch infection strategy when half of the available susceptible cells have been infected". Seeing as it is notoriously difficult for a virus to estimate what fraction of cells were infected, this raised a worry that persisted through reading the entire paper – until the very last paragraph, where it was finally discussed. My suggestions would be to do this (a) sooner, and (b) better.

We understand this concern and thank the reviewer for his/her detailed consideration and suggestions (discussed further below).

a) Sooner: I'd recommend explicitly commenting on the issue already in the Introduction and at least promise that it will be discussed.

We agree that it is better to address this point earlier in the manuscript. We have now further investigated how accurate the information carried by the arbitrium signal needs to be for communication to evolve (see point 1b), and have added a sentence to the Introduction referring to these new results: “Finally, we investigate how reliable the arbitrium signal needs to be for such communication to evolve, and find that the results are remarkably robust against variation in the density of bacteria”.

b) Better: “The arbitrium concentration during the early epidemic then is a direct reflection of the fraction of susceptible cells that have so far been infected”The readout is a direct reflection only of how many cells were lysed. In the model, yes, this is the same as fraction. But as the authors acknowledge at the very end this requires the assumption that bacterial cells were at carrying capacity K. In natural conditions, the carrying capacity can differ due to environmental factors – "unknown" to the virus.Far from being a minor worry, the issue is directly relevant for the authors' comparison between the communicating and the bet-hedging strategy. Under the communicating strategy, if the carrying capacity were modulated to e.g. half of its usual value, it seems that the virus would never switch to the lysogenic phase – a crippling blow to fitness. In contrast, the bet-hedging strategy seems vastly more robust to such a modulation. This suggests that the conclusions could be changed quite strongly if the carrying capacity were picked randomly (within some range) between passaging trials.It sounds like there is a prediction here.I do not intend this comment as negative / undermining authors' findings. Quite the opposite, I think by adding this analysis the authors could strengthen their argument, demonstrating the utility of a quantitative framework like theirs, and perhaps even make additional predictions. Beyond what level of (passage-to-passage) variability of K the bet-hedging strategy becomes favored? Addressing this comprehensively would be a project in itself; but generating a figure panel illustrating this in simulations would strengthen the authors' case for the benefits of modeling work.

This is a great suggestion, and we have now done such follow-up work. In these new simulations, at the start of each passaging episode a random value of the carrying capacity is sampled from a γ distribution. Sampling from a γ distribution is convenient for 2 reasons: (1) it allows us to change the variance of the distribution while maintaining the same mean, and (2) values sampled from this distribution are always >= 0. We performed these simulations for a long between-passage time (T = 24 h) and for varying levels of carrying capacity noise (*i.e.*, different values of the variance). The results of these new simulations are shown in a new Figure 5 and Figure 5—figure supplement 1.

Two things stand out when considering these new results. Firstly, a communication strategy with *φ*_max_ = 1 is selected for simulations in which the carrying capacity varies with a coefficient of variation (CV = standard deviation / mean) up to 0.35. The result that phages are selected that switch from a fully lytic to a fully lysogenic strategy is hence surprisingly resistant to noise in the carrying capacity. We do see that the response threshold value that is selected slightly declines with increasing levels of noise. This makes sense: as also stated by the reviewer, any passaging episodes in which the carrying capacity is lower than the phages’ response threshold are disastrous for the virus because under these conditions no lysogens are produced. As the variation in the bacterial carrying capacity increases, the phages hence become “more prudent”: they switch strategies earlier in each outbreak to avoid not switching at all in some outbreaks.

Secondly, the results partly confirm the hypothesis put forward by the reviewer, but with an important nuance. At high levels of variation in the carrying capacity (CV > 0.35), phages are selected with *φ*_max_ < 1. These phages hence show a bet-hedging strategy when the arbitrium concentration is above their response threshold value. Surprisingly, however, the response threshold value *θ* of the selected phages is never zero. We find this result for all levels of variation we tested, up to simulations in which the standard deviation on the carrying capacity is equal to its mean value (CV = 1). This suggests that even for very large variation in the bacterial carrying capacity a form of arbitrium communication is favoured over a completely bet-hedging strategy. Over the course of an outbreak, the selected phages still start with a lytic strategy, but do switch to a bet-hedging strategy at some low (but non-zero) arbitrium concentration.

To conclude, we find that the arbitrium communication system is more robust to variations in bacterial carrying capacity than we expected.

We included a short description of the set-up of these new simulations in the Materials and methods and discuss the results in a new section in the Results. Based on these new results, we also changed the corresponding paragraph in the Discussion.

An optional point:Since the cell-mediated uptake and degradation of arbitrium is also mediated by bacterial cells, and is thus also dependent on bacterial cell density, it's possible that this might mitigate the problem (making the arbitrium concentration less strongly dependent on K than naïve linearity). I don't know if it's true, but if it is, it would be cool, and worth highlighting. Unless I'm mistaken, this is not currently discussed. I think that could strengthen the story.

This is an interesting idea. However, for the parameter conditions used here it unfortunately does not seem to hold true (illustrated in Author response image 1). This has two reasons. First, the epidemiological dynamics tend to be very fast, and hence the production of arbitrium (due to infection events) overrules its uptake and degradation early in the epidemic. Second, during the active epidemic the number of cells temporarily declines because susceptible cells get lysed and their place is only later taken over by lysogens. Hence, the total uptake rate of arbitrium during the crucial active epidemic phase is relatively low due to the low number of cells.

Although we like the idea, we decided not to include it in the manuscript because the effect does not seem to occur in the current model set-up.

**Author response image 1. respfig1:** Short-term dynamics for lower values of the bacterial carrying capacity K.

2) Figure 3: In the competition between bet-hedging and communicating variants, what is the impact of initial conditions? Moreover, here, only the 2 top variants of each type are competed against each other. Is this sufficient to conclude that communicating variants will always be selected by competition? If binary comparisons are sufficient, this should be stated and explained. If not, then it would be important to show explicitly what happens when various phages of both types compete together.

The binary comparison shown in Figure 3C on its own is indeed not sufficient to conclude that communicating variants are always selected over bet-hedging variants. However, bet-hedging phage variants are also included in the large collection of possible variants in the simulations of Figure 3B. Strains with response threshold *θ_ι_* = 0 always have a lysogeny propensity of *φ*_max,*I*_, and hence do not respond to the arbitrium signal but rather employ a bet-hedging strategy. If any bet-hedging variant would be favoured over the communicating phages, it should hence dominate in Figure 3B. This does not happen, so from Figure 3B we can already conclude that the communicating strategy is favoured over the bet-hedging strategy. The one-on-one competition in Figure 3C was only added to underscore this conclusion.

To make clear that bet-hedging phages are included in the eco-evolutionary simulations with arbitrium communication, we now explain this in the Materials and methods. We have also rephrased part of the discussion of Figure 3B and 3C, to clarify that 3C is an example underscoring the conclusion already drawn from Figure 3B.

3) Scripts really should be openly available unless there is a compelling reason not to do so, especially in this case where the results depend so closely on the code. I would strongly urge the authors to instead publish all simulation scripts alongside the manuscript, both for archiving purposes but also to facilitate access.

We agree that publishing code is good practice, and apologise for initially taking the easy way out. We have now prepared the scripts for publication and made them available at github: https://github.com/hiljedoekes/PhageCom.

4) The parameters tested should be in the main text. Note that, other than the construction of the equations according to life history parameters, this is the only connection to "biology" that non-theoretical researchers will have to the manuscript. Table A1 should be in the main text. The table should list the conventional units of the parameters. For instance does burst size of 2 refer to 2 virions? This value would normally be in the 10's^-1^00's for most phages. Some of these parameters don't seem reasonable at first glance. For instance, the adsorption rate is usually something on the order of 10^-10^ to 10^-12^. I suspect this could be because these are "scaled" values. If the scaled values are made for model convenience they should also be listed separately. This way the connection between the model and the biology will be more evident to readers.

Thank you for pointing out that including the parameter values in the main text will help non-theoretical researchers relate to the model. This is something we had not considered. We have moved the parameter table from the appendix to the Materials and methods.

Indeed, the parameters originally presented in Table A1 were scaled parameters (which are unit-less). The scaled burst size (or “effective burst size”, as we call it) is a composite of the actual burst size *B* and the probability *b* that if a phage adsorbs to a susceptible cell, that cell actually gets infected.

Consequently, it is indeed lower than commonly known values of the burst size. The scaled adsorption rate is a composite of the “plain” adsorption rate, the reproduction rate of bacteria (such that time is measured in bacterial generations, rather than hours), and the carrying capacity of bacteria. As a result, it is indeed much higher than the commonly known adsorption rates per minute.

We understand that presenting the scaled parameters only might cause confusion. We have therefore extended Table 1 to now include both the estimates for the non-scaled parameters that we took from the literature, and the corresponding values for the scaled parameters as they were used in the parameter sweeps. This is now also briefly explained in the Materials and methods.

5) The idea of having to go away from a steady state for this to work is very appealing. It is also likely much closer to biological reality in this case. Other than models that specifically address serial passaging, is the approach of disturbing ESS's regularly a new concept? Is there a generalizable framework for this?

Thank you for this comment, we agree that the need to consider the system away from steady state is one of the key points in our work. We are not familiar with any works that describe a general framework for determining ESS’s in disturbed systems. While we appreciate the idea of developing such a general framework, we are hesitant to overstate our case. We expect that there are many systems (both in models and in the natural world) in which disturbances away from steady state drive evolution, but the exact nature of these disturbances and the way in which they affect selection pressures might differ from system to system. It is hence hard to imagine a general framework that goes beyond merely stating that “disturbances should be included in ESS calculations”, and we are currently unsure what such a general framework should look like.

6) Some conclusions should be specified or clarified:a) Figure 2 B-C: Some diversity seems to survive at steady state for T>5h. I believe this is what the authors refer to as "quasi-species" in the manuscript. It would be important to discuss and interpret this surviving diversity, and to explain whether it is going to survive forever or to slowly decay, and why.

This remaining diversity is indeed the viral quasi-species (Eigen, 1971). The quasispecies is shaped by the mutation-selection balance. While a single phage variant is the evolutionarily stable strategy (the most abundant variant, with *φ* = 0.04), phage variants with similar but slightly different *φ*-values constantly arise from mutations that occur during reproduction of the phages (*i.e.*, during the production of new phage particles in the lytic cycle). Because these mutants only differ slightly from the ESS, they are likely also only slightly less fit. Hence, selection against them is quite weak, and these mutants will linger for some time, also reproducing themselves (which yields additional mutants, that differ more from the ESS and hence experience stronger selection). Some diversity in the quasi-species will thus be maintained forever, and the level of this diversity is determined by a balance between influx of mutants because of mutations arising during the reproduction of the ESS-phage variant (mutation) and decline of the density of these mutants because they are outcompeted by the ESS-phage (selection). Only if we would remove mutations from our model (*i.e.*, set the mutation rate to zero) the diversity would decay until only the ESS-phage is left.

Since similar points were raised multiple times, we now realise that the quasi-species concept is not as widely known as we thought. We therefore included an explanation in our discussion of Figure 2B and C in the Results.

b) Figure 4: Please explain the difference between the analytical prediction and the simulation results at small values of bB in Figure 4B. In particular, does any of the simplifying assumptions result in this discrepancy?

The difference between the analytical prediction and the simulation at small values of *bB* indeed arises from violations of the simplifying assumptions we make to derive the ESS. The value of *bB* is the main determinant of the speed at which the epidemic unfolds. For small values of *bB*, the dynamics of the epidemic are slow. Then our simplifying assumption about the complete separation of time scales between the build-up of arbitrium during the epidemic and arbitrium uptake and degradation of by cells no longer holds. If the epidemic takes relatively long, the uptake and degradation of arbitrium by susceptible cells can no longer be ignored. As a consequence, the real arbitrium concentration at a certain time point (*i.e.*, the moment phages “should” switch from a lytic to a lysogenic strategy) will be lower than the arbitrium concentration used in the analytical approximation. We should hence expect that the response thresholds found in the simulations are lower than the analytical predictions. This is indeed the case.

We included a short version of this explanation in the Results.